# LEM4 confers tamoxifen resistance to breast cancer cells by activating cyclin D-CDK4/6-Rb and ERα pathway

Ang Gao[1], Tonghua Sun[1], Gui Ma[1], Jiangran Cao[1], Qingxia Hu[1], Ling Chen[2], Yanxin Wang[3], Qianying Wang[1], Jiafu Sun[1], Rui Wu[1], Qiao Wu[1], Jiaxi Zhou[4], Lin Liu[1], Junjie Hu [1,5], Jin-Tang Dong[1,6] & Zhengmao Zhu[1]

The elucidation of molecular events that confer tamoxifen resistance to estrogen receptor α (ER) positive breast cancer is of major scientific and therapeutic importance. Here, we report that LEM4 overexpression renders ER+ breast cancer cells resistant to tamoxifen by activating the cyclin D-CDK4/6 axis and the ERα signaling. We show that LEM4 overexpression accelerates tumor growth. Interaction with LEM4 stabilizes CDK4 and Rb, promotes Rb phosphorylation and the G1/S phase transition. LEM4 depletion or combined tamoxifen and PD0332991 treatment significantly reverses tamoxifen resistance. Furthermore, LEM4 interacts with and stabilizes both Aurora-A and ERα, promotes Aurora-A mediated phosphorylation of ERα-Ser167, leading to increase in ERα DNA-binding and transactivation activity. Elevated levels of LEM4 correlates with poorer relapse-free survival in patients with ER+ breast cancer undergoing endocrine therapy. Thus, LEM4 represents a prognostic marker and an attractive target for breast cancer therapeutics. Functional antagonism of LEM4 could overcome tamoxifen resistance.

[1] Department of Genetics and Cell Biology, College of Life Sciences, Nankai University, Tianjin 300071, China. [2] Department of Pathology, Tianjin Central Hospital of Gynecology and Obstetrics, Tianjin 300100, China. [3] Department of Pharmacology, Yong Loo Lin School of Medicine, National University of Singapore, Singapore, Singapore. [4] State Key Laboratory of Experimental Hematology, Institute of Hematology & Blood Diseases Hospital, Chinese Academy of Medical Sciences & Peking Union Medical College, 288 Nanjing Road, Tianjin 300020, China. [5] National Laboratory of Biomacromolecules, CAS Center for Excellence in Biomacromolecules, Institute of Biophysics, Chinese Academy of Sciences, Beijing 100101, China. [6] Department of Hematology and Medical Oncology, School of Medicine, Winship Cancer Institute, Emory University, Atlanta, Georgia. Correspondence and requests for materials should be addressed to J.H. (email: huj@ibp.ac.cn) or to J.-T.D. (email: j.dong@emory.edu) or to Z.Z. (email: zhuzhengmao@nankai.edu.cn)

The estrogen receptor (ER) pathway is considered an addictive oncogenic pathway in breast cancer cells. At least 70% of breast cancers are classified as ER+ breast cancers. Tamoxifen represents a mainstay adjuvant treatment in clinical practice over the past two decades. One-third of breast tumors that initially respond to the adjuvant therapy with tamoxifen will eventually relapse with endocrine-resistant disease[1]. The major mechanisms of endocrine resistance in ER+ breast cancers, through ERα itself, receptor tyrosine kinase (RTK) signaling, or cell cycle regulation with the cyclin D-CDK4/6-Rb pathway, have been demonstrated to be pivotal in endocrine therapy[2]. With regard to the cyclin D-CDK4/6-Rb pathway, the downstream or end points shared by multiple pathways including ERα signaling and RTK signaling could be targeted, which has the benefit of more directly targeting proliferation. The specific CDK4/6 inhibitor PD0332991 combined with endocrine therapy has been shown to substantially improve progression-free survival in patients with ER+ advanced breast cancer[3–5]. Although PD0332991 combined with endocrine therapy was approved as a first-line treatment for advanced ER+ breast cancer by the FDA (2015) and EMA (2016), no reliable biomarkers except ER status has been defined to diagnose tumors that depend on CDK4 activity and respond to CDK4/6 inhibitors[6].

Cancer cells often exhibit changes in nuclear morphology, and changes in nuclear morphology are a gold standard for clinical cancer diagnosis[7]. Breast cancer cells contain massive nuclear envelope (NE) invaginations[8]. Loss of NE integrity or NE rupturing, which results in genomic instability and uncontrolled exchange of nucleo-cytoplasmic content, may promote cancer progression[9–11]. However, very little is known about the mechanism by which disruption of the NE structure facilitates carcinogenesis and cancer progression. LEM proteins are the better-characterized NE proteins containing the LEM domain that interacts with the highly conserved essential chromatin-binding protein barrier-to-autointegration factor (BAF)[12]. LEM-BAF interactions form an important link between the NE and chromatin to maintain nuclear organization during interphase and in the timing of the post-mitotic NE reformation. LEM2 or LEM4 depletion resulted in nuclear shape defects[13,14]. Moreover, the highly dynamic localization and function of BAF during the cell cycle is tightly regulated by phosphorylation, which is temporally controlled by LEM4[13]. Based on these considerations, we hypothesized that some of the LEM proteins might function as oncoproteins, and any such role could be linked to dysregulation of the cell cycle machinery and activation of cyclin-dependent kinases.

Several studies investigating LEM proteins, including LAP2 and LEM3, have been reported in breast cancer[15,16]. In this study, we present evidence that LEM4 overexpression in ER+ breast cancer cells confers tamoxifen resistance through activation of both the cyclin D-CDK4/6-Rb pathway and the ERα signaling. By studying MCF7-TAMR cells and BT474 cells, we show that elevation of LEM4 expression is a key event to render ER+ breast cancer cells resistant to tamoxifen. LEM4 depletion or combined tamoxifen and PD0332991 treatment significantly overcomes the tamoxifen resistance. Moreover, LEM4 interacts with and stabilizes ERα, leading to increase in ERα DNA-binding and trans-activation activity. Therefore, LEM4 serves as a critical regulator in the transition of ER+ breast cancer cells to estrogen independence and tamoxifen resistance.

## Results

### LEM4 predicts clinical outcomes in breast cancer patients.
Breast cancer cells often exhibit massive NE invaginations[8,17]. In search of reasons that disruption of the NE structure would benefit a cancer cell, we interrogated the Cancer Genome Atlas database and found that LEM4, a member of the prominent family of NE proteins containing the LEM domain, was significantly overexpressed in breast tumors compared to normal breast epithelium (Fig. 1a, P < 0.001, Tukey's multiple comparisons test). To investigate the role of LEM4 in breast cancer, we performed immunohistochemistry (IHC) with commercial tissue microarrays (HBre-Duc150-Sur-01/02) and found that LEM4 was more highly expressed in tumor tissues from breast cancer patients and weakly detected in the paired noncancerous tissue regions (Fig. 1b, c). The IHC analysis also demonstrated that, in some cases, LEM4 was highly enriched in the nucleus of tumor cells (Fig. 1b), which is a surprising finding given that LEM4 has been shown to localize to the inner nuclear membrane and endoplasmic reticulum[13]. Moreover, cancer ATLAS analysis with an anti-LEM4 antibody revealed nuclear positivity in some breast cancer cases[18]. Next, we examined the Lem4 protein by IHC staining of the mouse mammary glands during four different stages (puberty, pregnancy, lactation, and involution). The results showed nuclear negativity in the mouse mammary epithelial cells (Supplementary Fig. 1a). Therefore, translocation occurred in the context of some cancer-related biological events. Although we are unable to reveal the mechanism that led to the nucleoplasm enrichment of LEM4, LEM4 expression significantly increased as tumors progressed to high-grade breast cancer (Fig. 1d). We then investigated whether LEM4 protein expression is associated with overall survival in 284 patients with breast cancer stratified according to breast cancer subtype and ER status. Patients were separated into two groups using the median expression of LEM4 as the dividing line and Kaplan–Meier survival analysis was performed. High LEM4 expression positively correlated with reduced overall survival (Fig. 1e). Patients with high LEM4 expression had greater overall decreased survival rate in luminal B and HER2-enriched breast cancer subtypes (Fig. 1f, g). In addition, in both ER+ patients and ER− patients, there was a significantly less chance of survival for patients with higher LEM4 expression (Fig. 1h).

We next performed a meta-analysis using an online Kaplan–Meier plotter breast cancer survival analysis to further assess the role of LEM4 in clinical outcomes (www.kmplot.com). We took advantage of the publically gene expression datasets from primary breast cancers with associated clinical data, including disease recurrence and survival (GSE2034[19], GSE2990[20], GSE16446[21], and GSE20685[22]). The results revealed that tumors with higher LEM4 expression had significantly worse relapse-free survival (Fig. 1i). In addition, patients with higher LEM4 expression had greater decreased relapse-free survival in both luminal A and luminal B subtype of breast cancers (Supplementary Fig. 1b). High LEM4 expression also positively correlated with worse overall survival in both ER+ patients and ER− patients (Supplementary Fig. 1c). Thus, increased LEM4 expression significantly correlates with decreased survival of patients with breast cancer.

### LEM4 expression promotes breast tumorigenesis.
To test the functional relevance of LEM4 overexpression in breast tumors, we stably expressed or depleted LEM4 in breast cancer cells and evaluated the cellular outcomes. Increased LEM4 expression in MCF7 cells (two clones: MCF7-LEM4 #1 and #2) enabled the cells to proliferate much faster than control cells (two clones: MCF7-vec #1 and #2) in monolayer culture as measured by SRB assay (Fig. 2a). Overexpression of LEM4 in T47D cells also resulted in increased cell growth (Supplementary Fig. 2a). Depletion of LEM4 from MCF7, T47D, BT474 or MCF7-LEM4 cells by RNA interference resulted in significantly decreased cell

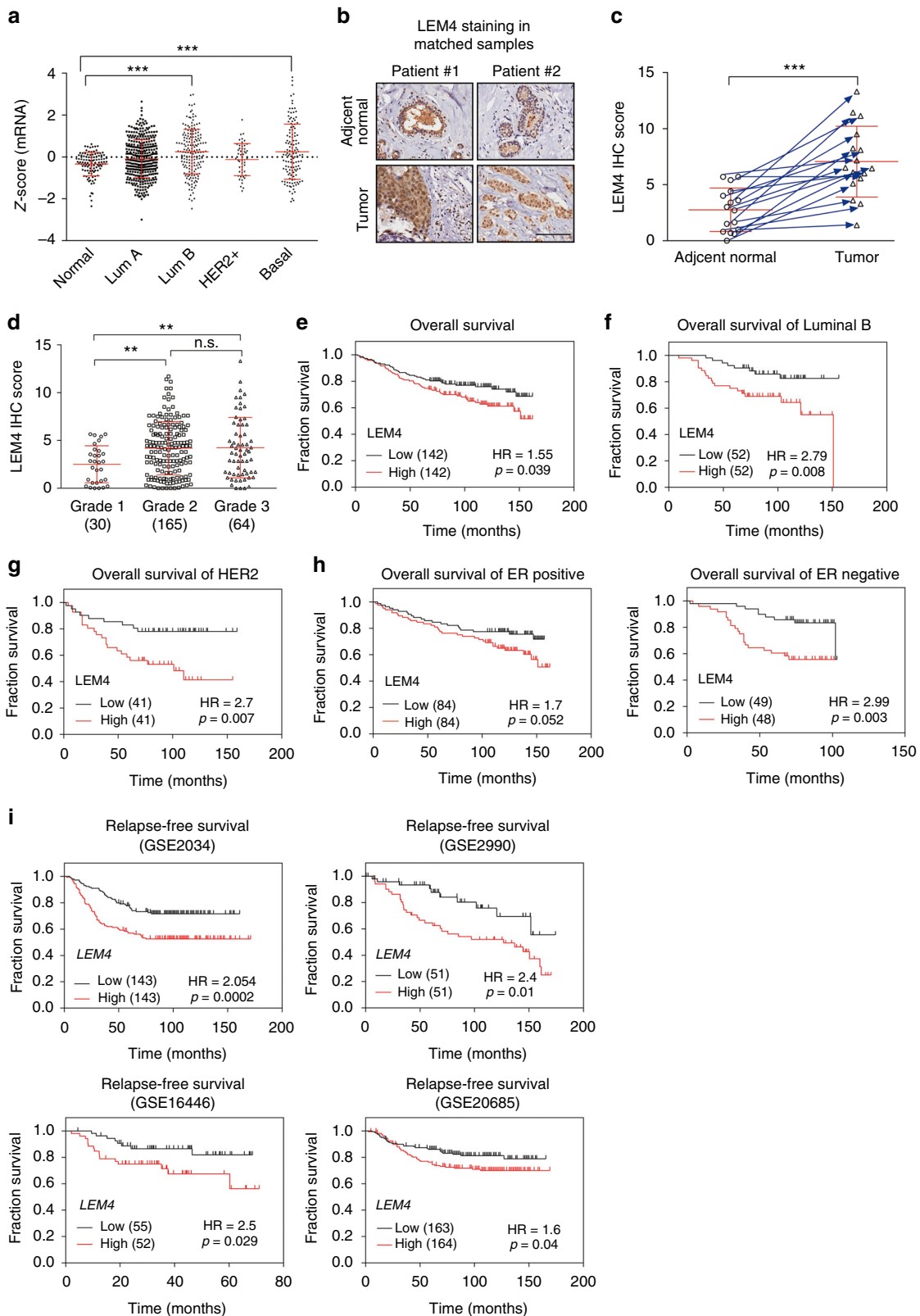

growth (Fig. 2b, c and Supplementary Fig. 2b, c). Similar results were observed in the non-tumorigenic epithelial cells MCF 10A, and RNA interference with *LEM4* expression resulted in a significant inhibition of mammosphere formation of MCF 10A cells in matrigel (Supplementary Fig. 2d, e). Evaluation of EdU

incorporation showed that elevated LEM4 in MCF7 cells or T47D cells gave rise to an increase in the number of EdU-positive cells (Supplementary Fig. 2f, g). Thus, LEM4 promotes cell proliferation in breast cancer cells and is necessary for cell proliferation in vitro.

**Fig. 1** High LEM4 expression correlates with poor survival of patients with breast tumors. **a** The Cancer Genome Atlas analysis showed the expression levels of LEM4 across different subtypes of breast cancer and normal tissues. Normal, $n = 106$, luminal A, $n = 372$, luminal B, $n = 174$, HER2, $n = 49$, basal-like, $n = 127$. **b** TMA analysis of LEM4 protein expression in breast cancers. Representative IHC images of LEM4 expression in matched normal and primary tumors from two patients are shown. Scale bars, 100 μm. **c** Scatter plot showing LEM4 protein expression in 16 paired normal and matched primary tumors, indicated by the blue arrows. **d** Scatter plots showing the LEM4 immunohistochemical staining results for 259 breast tumors in relation to cancer progression. **e–h** Kaplan–Meier analysis with median cutoff values of LEM4 expression and overall survival (**e**) in all 284 patients and patients who had luminal B or HER2 positive subtypes of breast cancer (**f**, **g**) or stratified according to ERα status (**h**). *P*-values were calculated by the log–rank test. **i** Kaplan–Meier analysis with median cutoff values of *LEM4* expression for breast cancer from GEO datasets. *P*-values were calculated by the log–rank test. **P < 0.01, ***P < 0.001, n.s., not significant. Tukey's multiple comparisons test for **a**, **d**. Paired *t*-tests for **c**

Since LEM4 is highly expressed in breast tumors and promotes cell proliferation, we hypothesized that LEM4 might enhance tumorigenesis. To investigate this speculation, we measured the ability of LEM4 to influence colony formation in soft agar. The results showed that MCF7-LEM4 cells had significantly increased colony numbers, whereas LEM4-depleted T47D cells yielded fewer colonies (Fig. 2d, e).

Next, we investigated whether LEM4 accelerates tumorigenesis in vivo with xenografts. MCF7 or T47D-derived cells were injected subcutaneously into athymic nude mice supplemented with a 60-day-release E2 pellet and tumor growth was monitored over time. Compared to MCF7-vec cells, MCF7-LEM4 cells formed faster growing and larger tumors (Fig. 2f). Furthermore, we observed that tumors originating from MCF7-LEM4 cells firmly attached to surrounding tissues with much greater proliferation ability, as indicated by immunostaining with an anti-Ki-67 antibody (Fig. 2h). LEM4-depleted T47D cells formed smaller tumors (Fig. 2g) with significantly lower expression of Ki-67 (Fig. 2i). Consistent with these findings, a positive correlation between *LEM4* and *MKI67* was observed at the mRNA level from the dataset GSE2990[20] ($r = 0.8544$) with statistical significance ($P < 0.0001$, Pearson's correlation test) (Supplementary Fig. 3). Given that MCF7-LEM4 cells grew as highly invasive tumors firmly attached to surrounding tissues, the MCF7-LEM4 cells were subjected to migration and invasion assays. We observed that MCF7-LEM4 cells were highly invasive in vitro (Supplementary Fig. 4a). Moreover, real-time RT-qPCR analysis showed that *Slug* and *ZEB1*, the epithelial-mesenchymal transition makers, were up-regulated in MCF7-LEM4 cells (Supplementary Fig. 4b). Western blot analysis showed that overexpression of LEM4 in MCF7 cells resulted in increased Slug expression (Supplementary Fig. 4c). Furthermore, immunostaining of MCF7-LEM4 cells using antibody (anti E-cadherin) showed the loss of E-cadherin in cell–cell contacts (Supplementary Fig. 4d). Thus, LEM4 overexpression promoted invasive and aggressive growth of MCF7-LEM4 cells.

One of the hallmarks capabilities of cancer is self-sufficiency in growth signals to sustain chronic proliferation[23]. MCF7 cells are estrogen-dependent for growth in vitro and in vivo, and although vector-transfected cells barely survived in estrogen-deprived medium, MCF7-LEM4 cells could grow in steroid-depleted medium (Supplementary Fig. 5a). In vivo, even in the absence of exogenous estrogen supplementation, the MCF7-LEM4 cells generated fast growing tumors with significantly higher expression of Ki-67 in athymic nude mice, whereas MCF7-control cells did not form palpable tumors (Supplementary Fig. 5b, c). Further, a time-course and dosage-course experiment revealed that LEM4 is not an estrogen-responsive gene (Supplementary Fig. 5d). Therefore, LEM4 overexpression enables MCF7 cells to be estrogen-independent for growth.

**LEM4 overexpression promotes the G1 to S phase transition.**
Dysregulated cell division, resulting in aberrant cell proliferation, is one of the key hallmarks of cancer. As LEM4 is a positive

regulator of cell proliferation in breast cancer cells, we performed a FACS analysis to address whether LEM4 promotes cell growth and enhances tumorigenesis via alteration of the cell cycle. Cell-cycle analysis revealed an increase in the number of cells in G1 phase and a decrease in the number of cells in S phase following depletion of LEM4 in T47D cells (Fig. 3a and Supplementary Fig. 6a). Similar results were observed in BT474 and MCF7 cells (Fig. 3b and Supplementary Fig. 6b, c). However, we observed that the proportion of cells in G1 phase was significantly decreased when LEM4 was overexpressed in MCF7 and T47D cells (Fig. 3c and Supplementary Fig. 6d, e). These data suggest that LEM4 alters the cell cycle by promoting the G1 to S phase transition.

Given that cell cycle progression was altered by modulating expression of LEM4, we sought to determine whether LEM4 regulates the expression of cell cycle-related genes. As LEM4 controls post-mitotic NE formation upon mitotic exit[13] and accelerates the G1/S phase transition, we focused on the genes for *CDK1*, *cyclin D*, *CDK4/6*, *cyclin E*, and *CDK2*, as well as *Rb* and *E2F1*. Real-time RT-PCR analysis indicated that the *CDK1*, *CDK2*, *cyclin D1*, *cyclin E1*, *Rb*, and *E2F1* mRNA levels decreased in LEM4-depleted T47D cells (Fig. 3d). In the MCF7-LEM4 cells, the *CDK1*, *CDK2*, *cyclin D1*, *cyclin E1*, and *E2F1* mRNA levels increased significantly (Fig. 3e). Western blot analysis showed that cyclin D1, CDK4, p-CDK4, Rb, and p-Rb decreased in the LEM4 depleted T47D and MCF7 cells (Fig. 3f, Supplementary Fig. 6f). Conversely, the level of cyclin D1, p-CDK4, Rb, p-Rb, E2F1, and cyclin E1 protein expression increased in the MCF7-LEM4 and T47D-LEM4 cells (Fig. 3g, Supplementary Fig. 6g). Consistent with these findings, the IHC analysis of tumors showed that cyclin D1, p-CDK4, and p-Rb exhibited a concerted upregulation in the MCF7-LEM4 xenografts and downregulation in T47D-shLEM4 xenografts (Fig. 3h, i). Thus, these data suggest that LEM4 regulates the expression of genes controlling the G1 to S phase transition.

**Overexpression of LEM4 renders cells resistant to tamoxifen.**
The gene expression signatures representing cell cycle progression can predict disease outcome in women treated with tamoxifen and suggests a possible mechanism for endocrine resistance[24]. Given that LEM4 overexpression enabled MCF7 cells to be estrogen-independent for growth, and the expression of CDK1, cyclin D1, CDK4/6, and CDK2, cyclin E were up-regulated in both MCF7-LEM4 and T47D-LEM4 cells, we sought to determine whether the LEM4 overexpression could account for tamoxifen resistance in ER+ breast cancers. We found elevated levels of LEM4 protein in MCF7-TAMR cells as compared to MCF7 cells (Fig. 4a). We then examined *LEM4* mRNA levels in the dataset GSE100075[25] from LTED models. The results revealed that *LEM4* expression was significantly elevated in MCF7-LTED models (Fig. 4b). In agreement with previous reports, tamoxifen alone had minimal effect on cell proliferation in MCF7-TAMR cells. However, siRNA knockdown of LEM4 was sufficient to inhibit cell proliferation with enhanced sensitivity to tamoxifen (Fig. 4c).

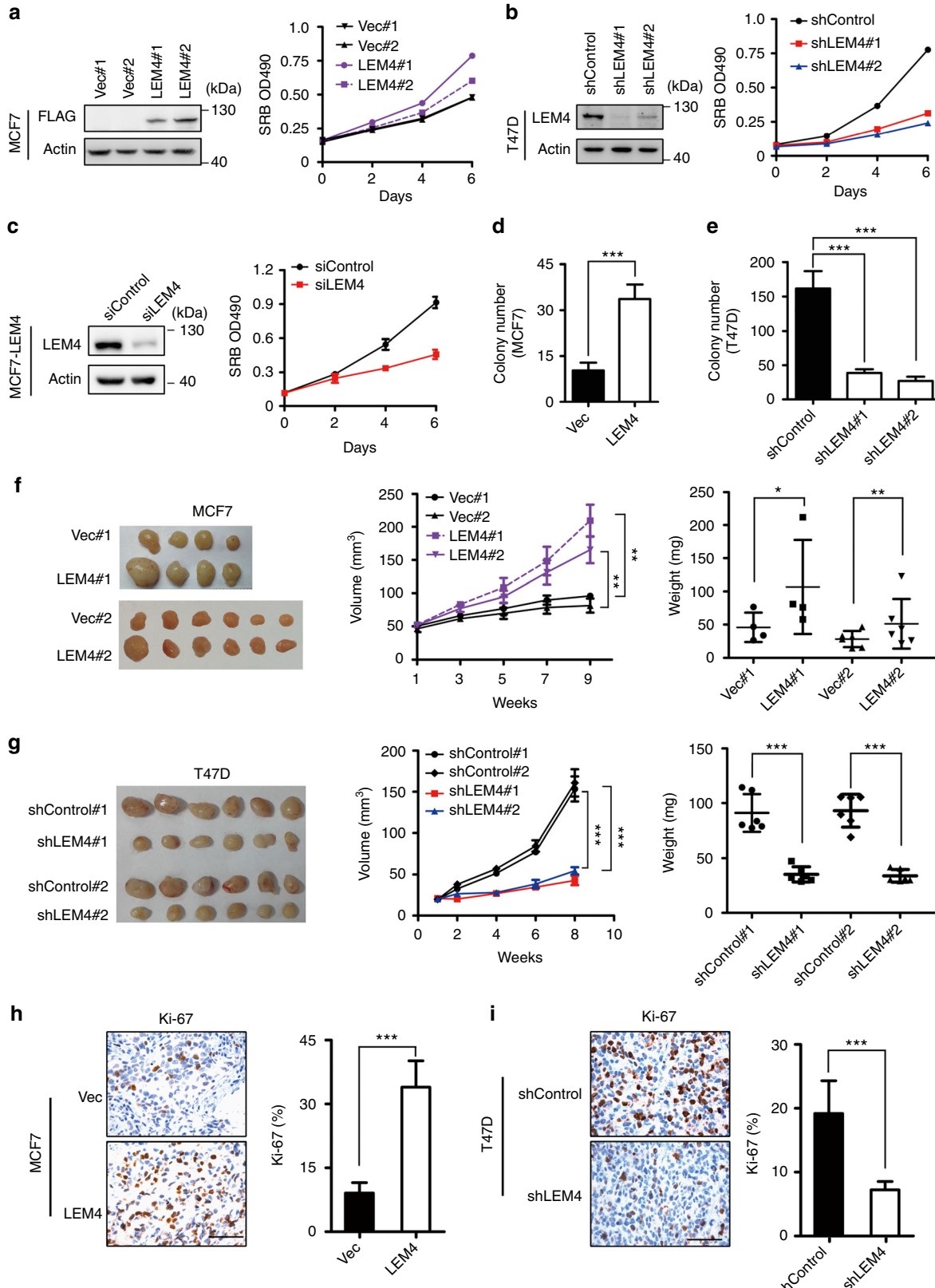

We then treated MCF7-LEM4 and MCF7 cells with various concentrations of tamoxifen and monitored cell survival. The dose-response curves showed that tamoxifen had much less effect on MCF7-LEM4 cell survival with IC50 values greater than 4 µmol L$^{-1}$ (Fig. 4d). Therefore, overexpression of LEM4 renders MCF7 cells resistance to tamoxifen.

BT474, a tamoxifen-resistant breast cancer cell line, is HER2 over-expression driving downstream signaling that leads to ligand

**Fig. 2** LEM4 promotes cell proliferation and tumorigenesis. **a** Growth curve of MCF7-vec and MCF7-LEM4 cells were measured by SRB assay in monolayer culture. Immunoblot was performed with anti-LEM4 antibody. **b** Growth curves of shControl-T47D, shLEM4#1-T47D, and shLEM4#2-T47D cells were measured by SRB assay in monolayer culture. Immunoblot was performed with anti-LEM4 antibody. **c** MCF7-LEM4 cells were transfected with LEM4 siRNA and control siRNA. Western blot was performed with anti-LEM4 antibody. Growth curves were measured by SRB assay in monolayer culture. **d**, **e** Soft agar colony formation by MCF7-vec and MCF7-LEM4 cells (**d**) or by T47D-shcontrol and T47D-shLEM4 cells (**e**). **f**, **g** Tumor growth of MCF7 and T47D cells implanted subcutaneously in athymic mice in the presence of an exogenous slow release estrogen implant. Mean ± s.e.m, $n = 4$ or 6 for MCF7 cells (**f**), $n = 6$ for T47D cells (**g**). **h**, **i** IHC for ki-67 in subcutaneous xenograft tumors from Figs. 2f, g. Mean ± s.d. for three independent replicates. Scale bars, 50 μm. *$P < 0.05$, **$P < 0.01$, and ***$P < 0.001$. Repeated measures ANOVA for **f** (volume), **g** (volume). Student's $t$-test for **f** (weight), **g** (weight), **h**, **i**

independent ERα activity. Given depletion of LEM4 in BT474 cells inhibited cell proliferation, we then investigated whether LEM4 overexpression is a key event in tamoxifen resistance through HER2 expression. We depleted LEM4 expression with *LEM4* siRNA in BT474 cells. The results revealed that LEM4 depletion did not alter the expression of HER2 (Supplementary Fig. 7a). Conversely, the LEM4 levels decreased upon knockdown of HER2 with *HER2* siRNA in BT474 cells (Supplementary Fig. 7a). Reduction of HER2 expression in BT474 cells by siRNAs enhanced sensitivity to tamoxifen (Supplementary Fig. 7b). In consistence with this finding, treatment of LEM4-depleted BT474 cells with tamoxifen resulted in significant cell death with IC50 values from greater than $4 \, \mu mol \, L^{-1}$ to $120 \, nmol \, L^{-1}$ (Fig. 4e). Thus, knockdown of LEM4 enhances tamoxifen anti-tumor effects in both MCF-TAMR and BT474 cells.

We next determined whether overexpression of LEM4 sufficed to induce tamoxifen resistance in vivo. Notably, unlike MCF7 cells, growth of MCF7-LEM4 cells as xenografts in immunodeficient mice failed to respond to the cytostatic/cytotoxic inhibition effects of tamoxifen (Fig. 4f). However, xenografts of BT474-shLEM4 cells regained sensitivity to tamoxifen and exhibited significant tumor regression (Fig. 4g). As LEM4 overexpression enabled MCF7 cells to be tamoxifen resistant, we investigated whether *LEM4* overexpression in primary breast tumors may prognosticate subsequent tamoxifen resistance. We analyzed the GEO datasets (GSE2990[20], GSE3494[26], and GSE9195[27]) of which the patients treated with adjuvant tamoxifen monotherapy (exclude all chemotherapy). We defined each dataset into two groups with respectively high and low level of LEM4. The Kaplan–Meier survival analysis results revealed that the group expressing high levels of LEM4 displayed a higher probability to develop recurrence as compared to the low group (Fig. 4h). Therefore, these data indicate that overexpression of LEM4 confers tamoxifen resistance.

**LEM4 activates the cyclin D-CDK4/6-Rb axis**. In culture, tamoxifen treatment leads to a G1 phase-specific cell cycle arrest and a consequence reduction in cell proliferation[28]. The actions of CDK4/6, through the phosphorylation of Rb, are pivotal in the transition from G1 to S phase in ER+ breast cancer cells[29]. Overexpression of LEM4 in MCF7 cells alters the phosphorylation of both CDK4 and Rb. In addition, analysis of BT474 cells as subcutaneous tumors treated with shRNA targeting LEM4 plus tamoxifen for 6 weeks showed significantly decreased expression of p-CDK4 and p-Rb (Supplementary Fig. 8a). To investigate whether PD0332991 was able to overcome the tamoxifen resistance induced by LEM4 overexpression, we treated MCF7-LEM4 cells and MCF7-TAMR cells with tamoxifen and PD0332991 alone or in combination and monitored cell survival. Combination treatment of cells resulted in significantly reduced cell growth in MCF7-LEM4 and MCF7-TAMR cells under estrogen-depleted conditions, as well as decreased p-Rb levels (Fig. 5a, b). Similar results were observed in BT474 cells (Supplementary Fig. 8b).

Next, we determined whether PD0332991 overcomes the tamoxifen resistance of MCF7-LEM4 cells in vivo. Tumor xenografts were established by injecting MCF7-LEM4 cells subcutaneously into athymic nude mice with estrogen supplementation. The mice were randomized to tamoxifen treatment, PD0332991 treatment or combined treatment until tumors reached an approximate volume of $100 \, mm^3$. The growth of MCF7-LEM4 cells remained unaffected by tamoxifen treatment alone as in the xenografts, but was suppressed by PD0332991, and the drug combination induced near-complete tumor regression (Fig. 5c). Analysis of tumors treated with PD0332991 plus tamoxifen for 6 weeks revealed reduced tumor cell density and increased fibrosis (Fig. 5d, H&E). Tumors treated with PD0332991 or the combination exhibited a decrease in Ki67+ tumor cells compared to the tamoxifen-treated tumors. Moreover, PD0332991 induced apoptosis (IHC analysis showed an increase in cleaved caspase-3/7-positive tumor cells with combined tamoxifen and PD0332991 treatment). p-CDK4 and p-Rb levels decreased similarly as in the xenografts of LEM4-depleted cells (Fig. 5d).

Given LEM4 is not a transcription factor and the role of LEM4 in the complex regulatory network modulating p-Rb function is unclear. We initially performed GST-pull down assays to test whether LEM4 binds to CDK4 and Rb. The results showed that GST-LEM4, but not GST, could pull-down CDK4 and Rb (Fig. 5e). We also performed co-immunoprecipitation (Co-IP) experiments in HEK293T cells following transfection of FLAG-CDK4 and GFP-LEM4 and found that GFP-LEM4 interacted with FLAG-CDK4 (Fig. 5f). In MCF7 cells, endogenous Rb was readily detected in FLAG-LEM4 immunoprecipitates (Fig. 5g). These data indicate that LEM4 binds to CDK4 and Rb. We further investigated whether loss of LEM4 results in CDK4 and Rb instability. We measured the half-life of CDK4 and Rb using a cycloheximide (CHX) chase assay. Degradation of both Rb and CDK4 was significantly aggravated at each time point in the LEM4-depleted cells (Fig. 5h). Overall, these data show that LEM4 enhances the stability and phosphorylation of Rb to promote the transition from G1 to S phase, resulting in tamoxifen resistance (Fig. 5i).

**LEM4 induces ERα transactivation activity**. Cyclin D1 and c-Myc were significantly upregulated in both MCF7-TAMR and MCF7-LEM4 cells (Fig. 6a). Remarkably, LEM4 knockdown reduced the expression of both cyclin D1 and c-Myc significantly in the two cell models (Fig. 6b). Cyclin D1 has been established as a major target of ERα, and cyclin D1 overexpression is associated with tamoxifen resistance[30,31]. To investigate whether LEM4 enhances ERα transactivation, MCF7 cells were transfected with LEM4 and ERE-luc, or ERα-negative MDA-MB-231 cells were transfected with LEM4, ERα, and ERE-luc. The reporter assay revealed LEM4 significantly enhanced ERα transactivation activity (Fig. 6c). To further explored the role of LEM4 in regulating the interaction of ERα with chromatin at the promoters of

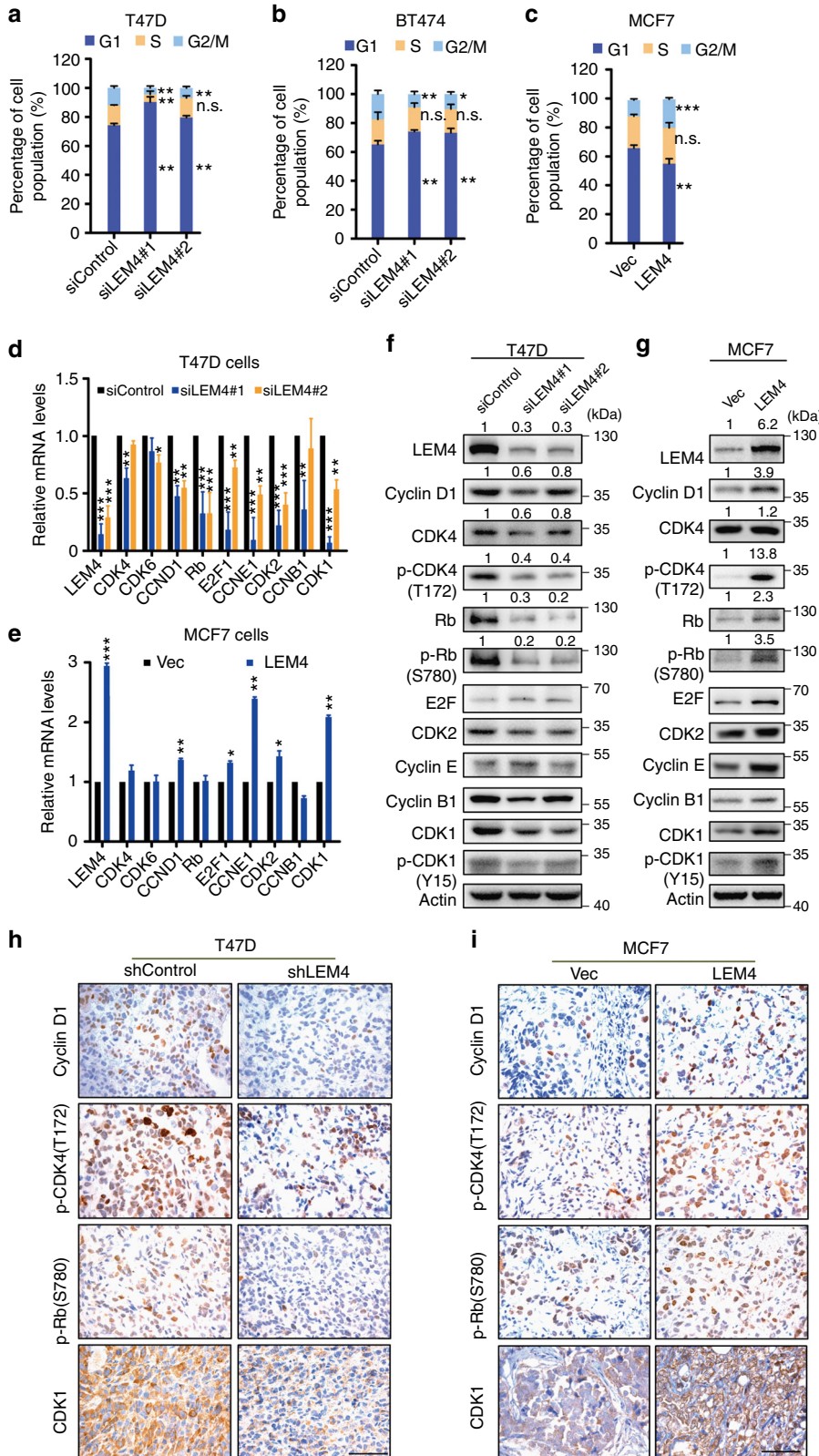

**Fig. 3** LEM4 overexpression promotes the G1 to S phase transition. **a–c** Depletion of LEM4 in T47D (**a**) and BT474 (**b**) cells and overexpression of LEM4 in MCF7 cells (**c**) altered the proportion of cells in G1, S, and G2/M phase by FACS analysis. **d, e** Real-time RT-PCR analysis of the cell cycle-related gene expression in T47D cells with LEM4-depleted (**d**) and LEM4-overexpressing MCF7 cells (**e**). **f, g** Immunoblot of cell cycle-related gene expression using the indicated antibodies in LEM4-depleted T47D cells (**f**) and MCF7-LEM4 cells (**g**). **h, i** Immunohistochemical analysis of the expression of cyclin D1, p-CDK4 (T172), p-Rb (S780), and CDK1 in tumors (Fig. 2f, g). Sizes of cell populations averaged from three independent experiments with standard deviations. Scale bars, 50 μm. *$P < 0.05$, **$P < 0.01$, *** $P < 0.001$. Tukey's multiple comparisons test for **a**, **b**, **d**, **e**. Student's t-test for **c**

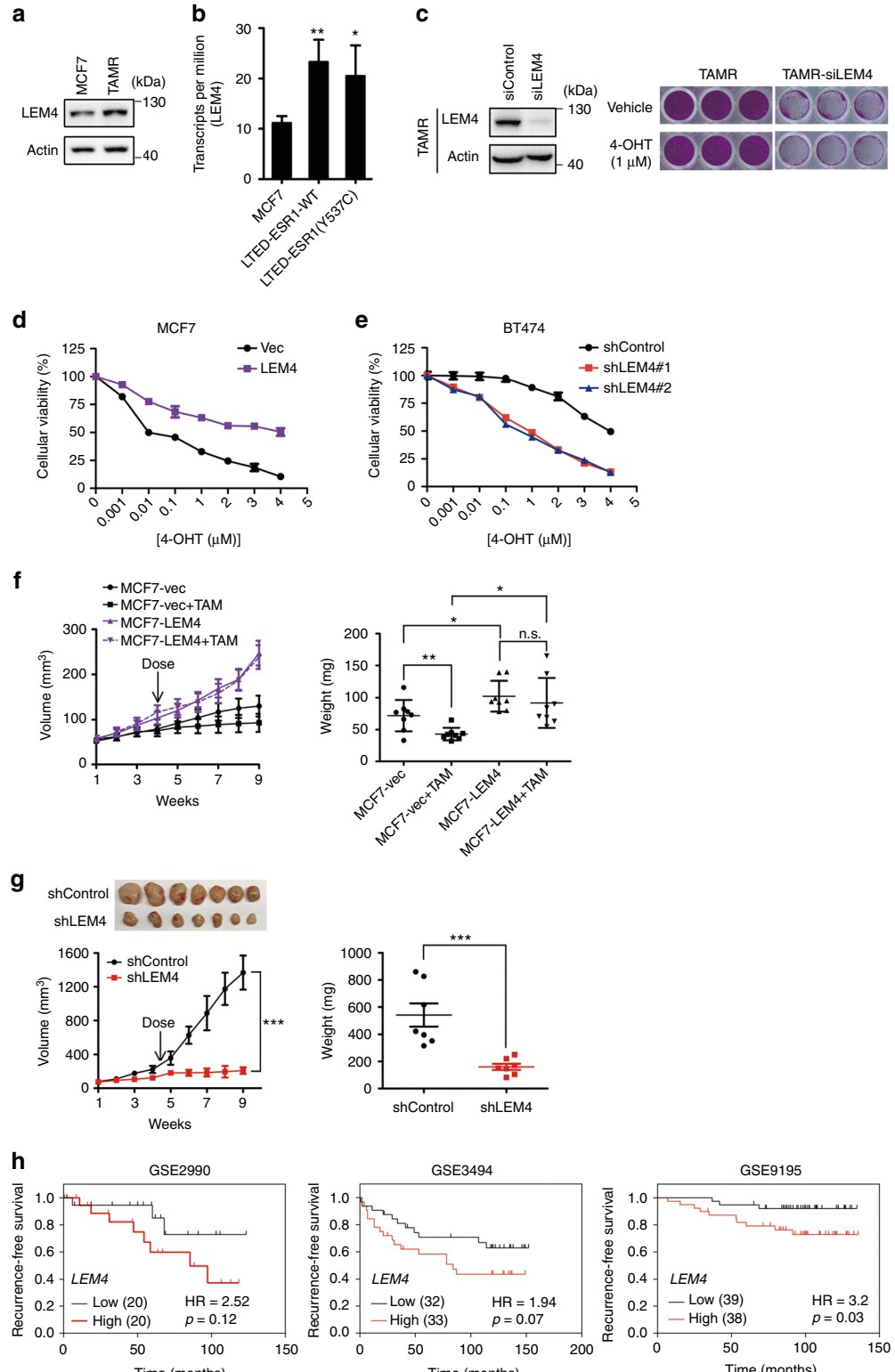

ERα target genes, we performed ERα chromatin immunoprecipitation (ChIP) analysis of known ERα-binding regions in the ERα target genes loci, *TFF1*, *PR*, *GREB1*, *CCND1*, and *c-Myc*. Following estrogen treatment for 45 min, the occupancy of ERα to the ERα-binding sites was significantly enhanced in MCF7-LEM4 cells (Fig. 6d). To gain more insight into the role of LEM4 on enhancing the recruitment of ERα at the promoters of ER target genes, a time course ChIP analysis was performed to

compare the kinetics of estrogen-stimulated loading of endogenous ERα at the promoter of ERα target genes. In MCF7 cells, ERα was recruited to the promoter of *PR* and *GREB1* in a dynamic fashion. In detail, ERα became rapidly bound to the promoter of *PR* and *GREB1*, within 15 min following E2 stimulation, the binding peaked by 30 min and had declined gradient by 60 min. While in MCF7-LEM4 cells, we observed a significant amount of ERα was present at the promoter of ER

**Fig. 4** LEM4 promotes tamoxifen resistance in ER positive breast cancers. **a** Western blot was performed with LEM4 antibody in MCF7 and MCF7-TAMR cells. **b** The GEO GSE100075 was downloaded from Gene Expression Omnibus (GEO).The expression level of *LEM4* in MCF7, MCF7-LTED-ESR1(WT), and MCF7-LTED-ESR1(Y537C) cells was measured by transcription per million (TPM). **c** *LEM4* siRNA and control siRNA treated MCF7-TAMR cells were treated with or without tamoxifen (1.0 μM) for 5 days. Total cell viability were assessed by SRB assays. Experiments were repeated three times, each experiment was triplicates. **d** A cellular viability assay was performed in MCF7-LEM4 and MCF7-vec cells treated with various concentrations of 4-OHT for 7 days. **e** BT474-shcontrol and BT474-shLEM4 cells were treated with various concentrations of 4-OHT for 7 days. Cellular viability was assessed by SRB assays. **f** Tumor growth and tumor weight of MCF7-vec and MCF7-LEM4 cells as subcutaneous xenografts in athymic mice with E2 supplementation until tumors reached 100 mm$^3$, then treated with or without tamoxifen (TAM) pellets implanted subcutaneously for 30 days. Mean ± s.e.m., $n = 8$. **g** Tumor growth of BT474-shControl and BT474-shLEM4 cells as subcutaneous xenografts in athymic mice with E2 supplementation until tumors reached ~200 mm$^3$, then treated with tamoxifen pellets implanted subcutaneously. Mean ± s.e.m., $n = 7$. **h** Three breast cancer datasets (GSE2990, GSE3494 and GSE9195) were from the KM Plotter database (www.kmplot.com). Kaplan–Meier analysis of recurrence-free survival in the cohorts of patients treated with adjuvant tamoxifen monotherapy (exclude all chemotherapy). Samples were stratified into "high" and "low" *LEM4* expression based on median cutoff value in each dataset. *P*-values were calculated by the log–rank (Mantel–Cox) test. n.s., not significant, *$P < 0.05$, **$P < 0.01$ Tukey's multiple comparisons test for **b**, **f** (weight). Repeated measures ANOVA for **g** (volume), Student's *t*-test for **g** (weight)

target genes, and the signal of ERα at the promoter was relatively constant until later time points. Strikingly, a less amount of the signal of ERα (at the promoter of *CCND1* and *GREB1* gene) or no signal of ERα (at the promoter of *PR* gene) was detected at the same locus until late time points in the LEM4-depleted MCF7-LEM4 cells (Fig. 6e). Therefore, these data suggest that LEM4 activates ERα transactivation activity and that LEM4-induced ERα activation could not be inhibited by tamoxifen.

**LEM4 interacts with and stabilizes ERα**. The mechanisms by which LEM4-induced ERα transactivation activity is an interesting question. We observed a higher level of ERα and p-ERα-Ser167 in both MCF7-LEM4 and MCF7-TAMR cells compared to the control MCF7 cells (Fig. 6a). Conversely, LEM4 knockdown reduced both ERα and p-ERα-Ser167 levels (Fig. 6b). Similar results were observed in LEM4-depleted BT474 cells (Supplementary Fig. 9a). In agreement with these findings, we observed an elevated abundance of p-ERα-Ser167 in MCF7-LEM4 xenografts and decreased abundance in BT474-shLEM4 xenografts (Supplementary Fig. 9b). Understanding that human ERα is rapidly degraded in mammalian cells in an estradiol-dependent manner[32]. We investigated whether LEM4 could prevent ERα degradation. MCF7-LEM4 cells were treated with E2 for 30 min. Interestingly, we observed that ERα was not degraded in MCF7-LEM4 cells (Fig. 7a). To evaluate whether LEM4 reduction is related to ERα stability, we measured the half-life of ERα using a CHX chase assay. As shown in Fig. 7b, degradation of ERα was accelerated at each time point in LEM4-depleted cells. These data suggested that LEM4 might interact with ERα to prevent ERα degradation. As shown in Fig. 7c, LEM4 was detected in ERα immunoprecipitates in BT474 cells. Furthermore, Co-IP and GST pull-down assays revealed a direct interaction between LEM4 and ERα, and the interaction between ERα and LEM4 occurred at the DNA-binding domain conclude Serine-167 (Fig. 7d, e). Moreover, immunofluorescence staining for ERα and LEM4 in MCF7-LEM4 (FLAG tagged) cells showed that ERα co-localized with LEM4 not only in the NE but also in cytoplasm (Fig. 7f). Thus, LEM4 physically interacts with and stabilizes ERα.

In addition, we determined whether unliganded ERα is required for the estrogen-independent growth of MCF7-LEM4 cells. MCF7-LEM4 cells were transfected with siRNA of *ESR1* or treated with fulvestrant for 6 days, we found that downregulation of ERα inhibited estrogen-independent growth of MCF7-LEM4 cells (Fig. 7g). A further exploration in dataset GSE33658, which was designed for a phase II neoadjuvant trial of anastrozole (A), fulvestrant (F) and gefitinib (G) in patients with newly diagnosed ER+ breast cancer[33]. As shown in Fig. 7h, at the post-treatment in both AF-treatment and AFG-treatment group, *LEM4* mRNA level reduced in the patients with complete

response or the partial response disease-state, while the expression level of *LEM4* increased in the patient with progressive disease-state.

**LEM4 mediates the phosphorylation of ERα-Ser167 by Aurora-A**. Phosphorylation of ERα-Ser167 has been shown to sufficiently upregulate cyclin D1[34–36]. Given the phosphorylation level of ERα-Ser167 was significantly altered when LEM4 was overexpressed or depleted in ER+ breast cancer cells, we then determined whether LEM4 could directly regulate the phosphorylation of ERα–Ser167. ERα-Ser167 has been shown to be phosphorylated by Aurora-A, AKT, and S6K1[34–36]. We observed that only Aurora-A increased in the MCF7-LEM4 cells, and a large amount of p-Aurora-A was induced in MCF7-LEM4 and MCF7-TAMR cells (Fig. 8a). Whereas both Aurora-A and p-Aurora-A decreased significantly in LEM4-depleted MCF7-TAMR cells or the LEM4-aboragated MCF7-LEM4 cells (Fig. 8b). Similar results were observed in BT474-shLEM4 cells (Fig. 8c). Next, we assessed the levels of p-ERα-Ser167 in MCF7-LEM4 cells treated with siRNAs against *AKT*, *Aurora-A*, and *LEM4* respectively. Immunoblot analysis showed that LEM4 or Aurora-A depletion resulted in a great reduction in p-ERα-Ser167 (Fig. 8d). Given that Aurora-A interacts with and phosphorylates ERα[36], we determined whether LEM4 actively interacts with Aurora-A to phosphorylate ERα. Co-IP revealed that LEM4 interacts with Aurora-A (Fig. 8e). Furthermore, we performed Co-IP in LEM4-depleted HEK293T cells following the transfection of FLAG-ERα and GFP-Aurora-A. The results demonstrated that depletion of LEM4 decreased the interaction between ERα and Aurora-A in vivo (Fig. 8f). Moreover, the CHX chase assay revealed that LEM4 contributes to the stability of the Aurora-A protein in both MCF7 cells and BT474 cells (Fig. 8g). Thus, LEM4 enhances Aurora-A-mediated phosphorylation of ERα on Ser167 and promotes ERα mediated transcription of *CCND1* and *c-Myc* (Fig. 8h).

## Discussion

Endocrine therapy is the cornerstone of treatment for patients with ER+ breast cancer[37,38]. However, the emergence of resistance to long-term endocrine treatment is inevitable in a proportion of patients with advanced breast cancer. The major challenge for successful treatment remains to identify new therapeutic targets or more specific biomarkers that are predictive of the therapeutic responses to endocrine therapy. Here, we characterize a critical role of LEM4 overexpression in tamoxifen resistance. Firstly, LEM4 accelerates malignant cell growth and breast tumorigenesis. Moreover, the overexpression of LEM4 enables MCF7 cells to be estrogen-independent for growth.

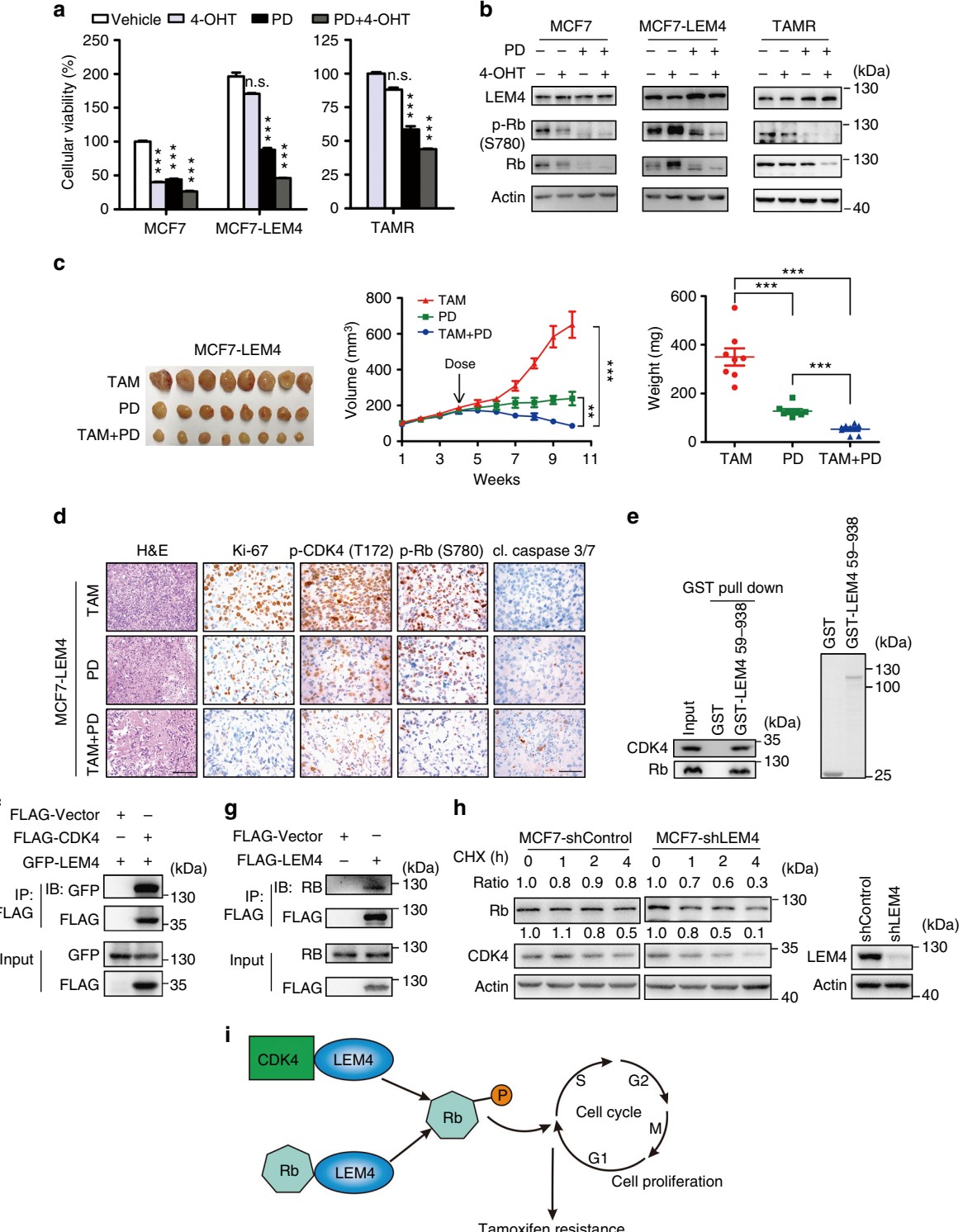

Secondly, LEM4 alters the cell cycle by promoting the G1/S phase transition. Cyclin D1, p-CDK4, and p-Rb exhibit a concerted upregulation in the LEM4 overexpressing ER+ breast cancer cells. Overexpression of LEM4 renders MCF7 cells resistant to tamoxifen, and siRNA knockdown of *LEM4* or combination treatment with PD0332991 significantly overcome tamoxifen resistance among the MCF-TAMR, BT474, and MCF7-LEM4 cells. Thirdly, LEM4 not only stabilizes ERα via interaction with ERα, but also induces ERα transactivation activity. Moreover, LEM4 enhances Aurora-A-mediated phosphorylation of ERα on

Ser167. The fourth, elevated expression of LEM4 correlates with poor survival of patients with breast tumors. Data mining analysis of several GEO datasets with breast cancer patients who received systemic endocrine therapy revealed that a higher level of LEM4 was associated with poorer recurrence-free survival. Thus, LEM4 appears to be a major causal factor in endocrine therapy resistance.

Long-term endocrine treatment often leads to acquired resistance in ER+ breast cancer. Data indicate that this may be mediated by multiple mechanisms that can potentiate cyclin

**Fig. 5** LEM4 confers tamoxifen resistance by activating the cyclin D-CDK4/6-Rb axis. **a** MCF7, MCF7-LEM4, and MCF7-TAMR cells were treated with 5% DCC-FBS (vehicle), 4-OHT (1 μM), PD0332991 (PD) (0.2 μM), or a combination of 4-OHT and PD0332991. Adherent cells were tested by SRB after 9 days. Data are presented as % parental control. Mean ± s.d. for three independent replicates. **b** Immunoblots of lysates from cells treated as in **a** with indicated antibodies. **c** Tumor growth of MCF7-LEM4 cells as subcutaneous xenografts in athymic mice with E2 pellets when tumors reached an approximate volume of 100 mm$^3$, then treated with tamoxifen pellet implanted subcutaneously, 100 mg kg$^{-1}$ PD0332991 (tricubic weekly), or a combination of tamoxifen pellet and PD0332991. Mean ± s.e.m., $n = 8$. **d** H&E staining and IHC for Ki-67, p-CDK4, p-Rb, and cleaved caspase 3/7 from **c**. Scale bars for H&E, 150 μm. Scale bars for IHC, 50 μm. **e** GST alone or recombinant GST-LEM4 immobilized on glutathione-agarose beads was incubated with the MCF7 cell extract. The pulled-down proteins were analyzed by immunoblotting with CDK4 and Rb antibodies. **f** HEK293T cells were transfected with GFP-LEM4 and pCMV6-FLAG-CDK4 or the empty vector pCMV6. The interaction of FLAG-CDK4 with GFP-LEM4 was analyzed by immunoprecipitation of the cell lysate with anti-FLAG affinity gel and immunoblotted with anti-GFP antibody. **g** HEK293T cells were transfected with pCMV6-FLAG-LEM4 or the empty vector pCMV6. The interaction of FLAG-LEM4 with Rb was analyzed by immunoprecipitation of the cell lysate with anti-FLAG affinity gel and immunoblotted with Rb antibody. **h** MCF7-shControl and MCF7-shLEM4 cells were treated with 50 μg mL$^{-1}$ CHX for 0, 1, 2, and 4 h and Western blotting was performed. **i** Model of LEM4 regulation of the cyclin D-CDK4/6-Rb axis leading to tamoxifen resistance in ER+ breast cancer. n.s., not significant. \*\*$P < 0.01$, \*\*\*$P < 0.001$. Tukey's multiple comparisons test for **a**, **c** (weight). Repeated measures ANOVA for **c** (volume)

D1-CDK4/6-Rb signaling in an ERα-independent manner[39–42]. Overexpression or amplification of both cyclin D1 and CDK4 is especially high in the luminal B (58% and 25%, respectively) and HER2-enriched subtypes (38% and 24%, respectively)[42]. Consistent with these previous findings, our results reveal that patients with higher LEM4 expression have an even greater decrease in overall survival for luminal B and HER2-enriched subtypes of breast cancer. Further, LEM4 functions via a simultaneous increase in the protein levels of cyclin D1, p-CDK4, and p-Rb, each of which are reversed in LEM4-depleted cells. CDK4 activation requires both binding to cyclin D1 and its phosphorylation on Thr172[43,44]. CDK4-Thr172 phosphorylation most strongly correlates with sensitivity to PD0332991[43]. We show that PD0332991 treatment results in a complete response to sensitizing MCF7-LEM4 cells to tamoxifen treatment. Furthermore, knockdown of LEM4 not only correlated with decreased p-CDK4 and p-Rb, but also restored tamoxifen sensitivity to both MCF7-TAMR and BT474 cells. This functional overlap prompted our hypothesis that LEM4 acts as an A-kinase anchor protein to activate the cyclin D-CDK4-Rb signaling axis. GST-pull down assays and Co-IP studies directly support LEM4 interactions with CDK4 or Rb. The CHX chase assay suggests that LEM4 is required for CDK4 or Rb protein stabilization. Given LEM4 could coordinate the activities of VRK1 and PP2A to enable NE reassembly during mitosis[13], we immunoblotted VRK1 and PP2A-C in the LEM4-depleted MCF7 cells. The results revealed that VRK1 decreased but PP2A-C elevated, and VRK1 protein degradation was independent on LEM4 (Supplementary Fig. 10a, b). VRK1 kinase has been identified as a marker for a subgroup with a poorer prognosis within the ER+ cases[45–47]. While activation of PP2A-C is required for E1A-mediated sensitization to drug-induced apoptosis[48]. Therefore, the role of LEM4 to prevent protein degradation would be target-protein dependent. Here, our data implicate LEM4 as a key regulator of the cyclin D-CDK4-Rb axis that promotes the G1/S phase transition.

Cyclin D is a direct transcriptional target of ERα[49], and overexpression of cyclin D has been implicated in tamoxifen resistance[50–52]. Our findings showed that LEM4 induced ERα transactivation activity. Previous studies have shown that loss of ERα expression occurs in only a minority (15–20%) of resistant breast cancers[40]. Here we show that LEM4 depletion aggravates ERα degradation. Overexpression of LEM4 prevents the rapid degradation of ERα in mammalian cells in an estradiol-dependent manner. Furthermore, ERα downregulation inhibits estrogen-independent growth of MCF7-LEM4 cells. Thus, ERα signaling might render the cell more dependent on LEM4-mediated pathways such as ERα itself. It is worthwhile to note that phosphor-ERα-Ser167 is one of the major mechanisms causing tamoxifen resistance[34,36,53]. Moreover, phosphor-ERα-Ser167 has been

demonstrated to sufficiently upregulate cyclin D[36]. Our data show that LEM4 induces elevations of both Aurora-A and phosphor-Aurora-A. Aurora-A has been shown to promote distant metastases only in ER+ breast cancer cells and renders breast cancer cells resistant to tamoxifen[36,54,55]. The increased stability of Aurora-A protein results in elevated phosphor-ERα-Ser167 levels. Collectively, these findings provide an underlying mechanism for LEM4 activation of ERα signaling and contribution to increased cyclin D1 expression. In addition, overexpression of HER2 is one of the best characterized mechanisms of endocrine resistance, our data implicate LEM4, might like emerin[56], act as a downstream effector for HER2 signaling pathway. Exploration results of dataset GSE33658, which based on a phase II neoadjuvant trial of anastrozole (A), fulvestrant (F), and gefitinib (G) in patients with ER+ breast cancer, showed that decreased expression of LEM4 was associated with complete response or partial response to both AF-treatment and AFG-treatment. Thus, functional antagonism of LEM4 might allow the attack of multiple therapeutic targets simultaneously in breast cancer.

In summary, our study presents a more integrated visual of how LEM4 proteins orchestrate the major group of molecules controlling the cyclin D-CDK4-Rb axis activated during the G1/S phase transition. Moreover, we present evidence for LEM4 acting as a scaffold for both Aurora-A and ERα and promoting the activation of ERα signaling. The activated cyclin D-CDK4/6-Rb signaling and ERα signaling subsequently drive the transition of breast cancer cells to estrogen independence and tamoxifen resistance. Further study is needed to characterize the functions of the interaction of LEM4 proteins with mitotic kinases, such as CDK1, Aurora-A, and Aurora-B, during tumorigenesis and metastasis.

## Materials and methods

**Cell lines and cell culture.** The human breast cancer cell lines MCF7, T47D, BT474, and MDA-MB-231 were purchased from American Type Cell Culture (Manassas, VA) and cultured in DMEM or RPMI-1640 medium supplemented with 10% fetal bovine serum (FBS). The MCF7-TAMR cell model was kindly provided by Dr Tao Zhu (University of Science and Technology of China)[57,58]. All cells were maintained at 37 °C with 5% CO$_2$. For deriving vector-control and LEM4-overexpression cell lines, pCMV6 vector or pCMV6-3×FLAG-LEM4 was stably transfected into MCF7 and T47D cells using Lipofectamine 2000 (Invitrogen, Thermo Fisher Scientific). To generate LEM4 shRNA cells, two different shRNA hairpins specifically targeting human LEM4 (Supplementary Table 1) were cloned into LKO.1 and used to knock down LEM4 constitutively in various breast cancer cell lines. All cell lines had been authenticated by STR profiling analysis.

**Construction of expression plasmids.** The pCMV6-FLAG-LEM4 and pCMV6-FLAG-CDK4 plasmids were purchased from OriGene. Mammalian expression plasmids for GFP-Aurora-A and GFP-ERα were generated in our laboratory. The mammalian expression plasmids for various ERα mutants were constructed by PCR amplification using the primers listed in Supplementary Table 2. The bacterial

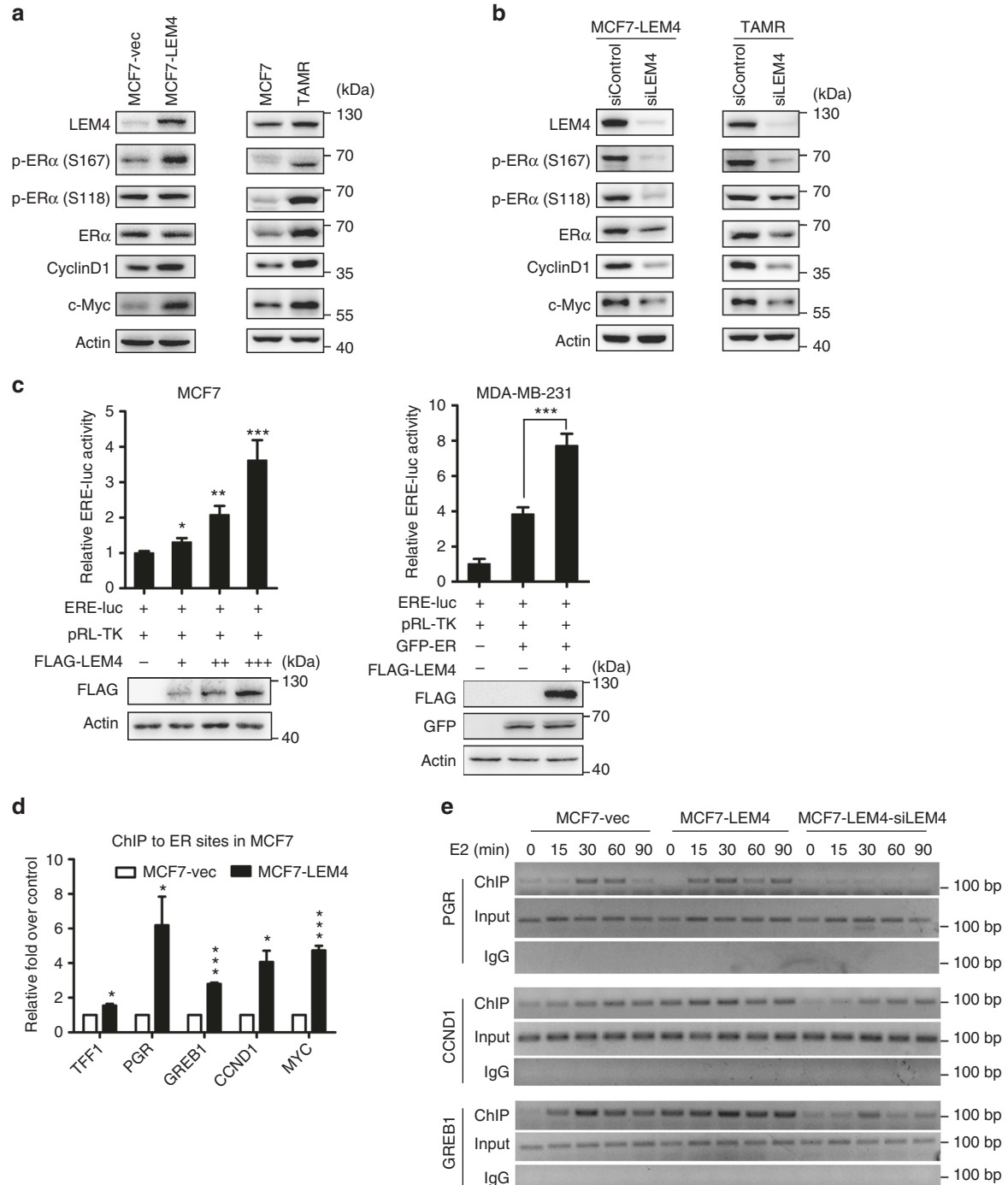

**Fig. 6** LEM4 induces ERα transactivation activity. **a** Immunoblot analysis of the phosphorylation of ER, Cyclin D1, and c-Myc in MCF-LEM4 and MCF7-TAMR cells. **b** Immunoblot analysis of the phosphorylation of ER, Cyclin D1, and c-Myc in LEM4-depleted MCF7-LEM4 and LEM4 knocked-down MCF7-TAMR cells. **c** Luciferase assay. ER+ MCF7 and ERα-negative MDA-MB-231 cells were transfected with ERE-Luc and other indicated plasmids. Following incubation for 48 h, luciferase activity was measured and normalized to Renilla. Results are the mean ± s.e.m. of three independent experiments performed in triplicate. **d** ERα ChIP assay of known ER-binding sites in ERα target genes was performed in MCF7-LEM4 cells incubated in estrogen-depleted medium (5% charcoal-stripped serum in phenol red-free DMEM) for 72 h before treatment with vehicle, 10 nmol L$^{-1}$ E2 for 45 min. **e** Time course ChIP study of the endogenous ERα with the estrogen response elements in the promoter region of the ERα target genes. MCF7, MCF7-LEM4, and LEM4-depleted MCF7-LEM4 cells incubated in estrogen-depleted medium (5% charcoal-stripped serum in phenol red-free DMEM) for 72 h before treatment with vehicle, 10 nmol L$^{-1}$ E2. ChIP analysis was conducted by using anti-ERα antibody. *$P < 0.05$, **$P < 0.01$, ***$P < 0.001$. Tukey's multiple comparisons test for **c**. Student's *t*-test for **d**

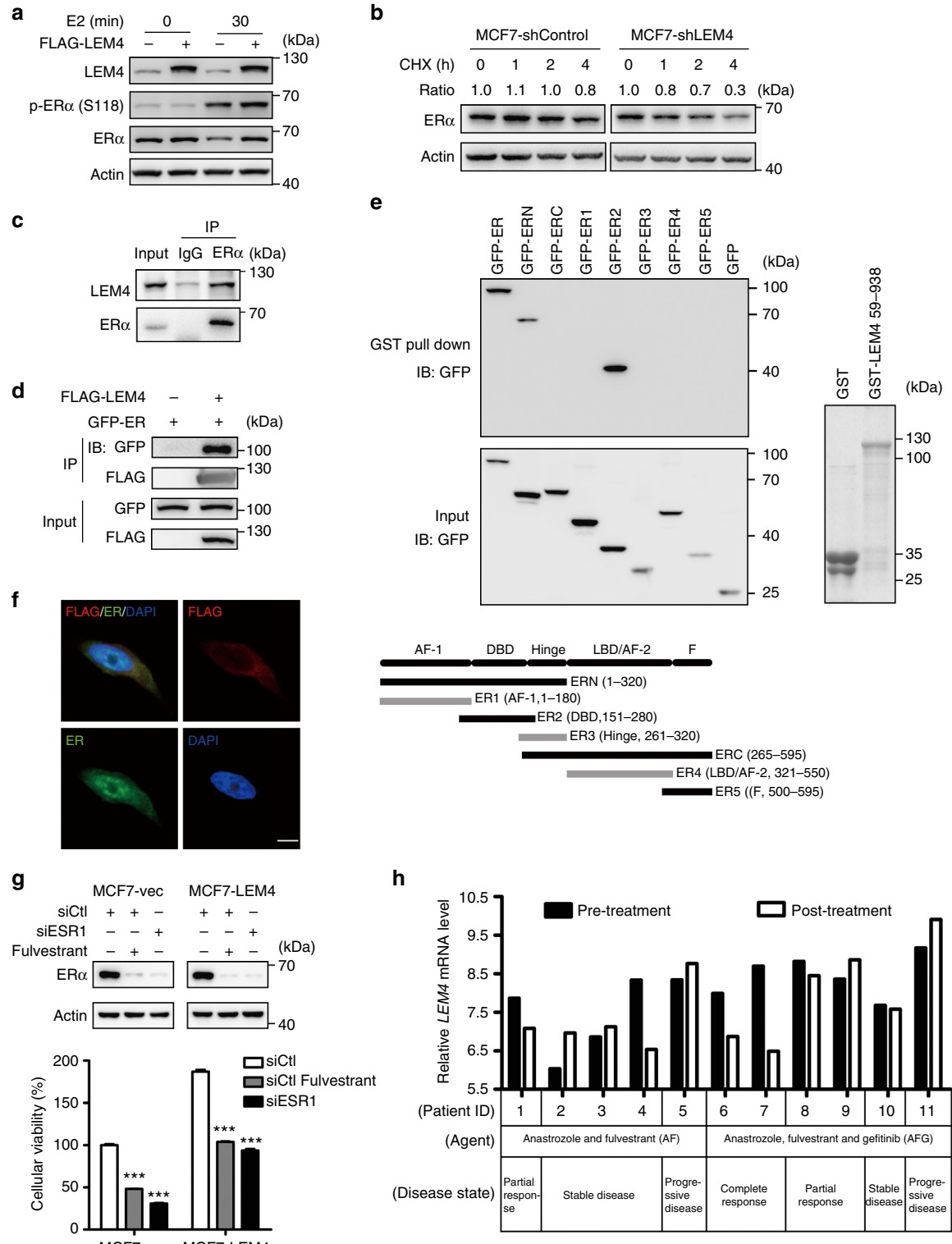

expression plasmids for GST-LEM4 (containing amino acids 59–938) were constructed by insertion of the cDNA fragments into the pGEX-6P-1 vector.

**Breast cancer molecular subtype and survival analysis.** Cancer subtype-specific LEM4 gene expression analysis was performed on TCGA_BRCA_exp_HiSeqV2-2015-02-24 mRNA expression data. The mRNA-scores were normalized and expressed as a $Z$-scores following the formula $Z = (LEM4\_score$ of each sample – mean LEM4_score)/(s.d. of all samples). Kaplan–Meier survival analysis was performed using an online database (www.kmplot.com) and the data was analyzed

using median cutoff in each case. The study survival curves based on the LEM4 protein expression score in 284 patients were plotted using Kaplan–Meier analysis and the statistical parameters calculated by log–rank (Mantel–Cox) test using GraphPad Prism 6 software.

**Cell proliferation and viability assays.** Cell proliferation was monitored by SRB assay and EdU (5-ethynyl-2′-deoxyuridine) incorporation assay. The SRB assay was performed following an already established procedure[59]. Briefly, cells were plated into 24-well plates, followed by incubation of cells with treatment of

**Fig. 7** LEM4 interacts with and stabilizes ERα. **a** Immunoblot analysis of ERα in MCF7 and MCF7-LEM4 cells grown under estrogen-deprived conditions in DMEM phenol-free medium containing 5% dextran charcoal-stripped serum and treated with 10 nmol L$^{-1}$ E2 for 30 min. **b** MCF7-shControl and MCF7-shLEM4 cells were treated with 50 μg mL$^{-1}$ CHX for 0, 1, 2, and 4 h and ERα analyzed by immunoblot. **c** For endogenous LEM4 and ERα interaction, BT474 cells were immunoprecipitated with anti-ERα antibody and detected with anti-LEM4 antibody. **d** HEK293T cells were transfected with FLAG-LEM4 and GFP-ERα or the empty vector pCMV6. After incubation for 48 h, cell lysates were precipitated with anti-FLAG affinity gel and immunoblotted with anti-GFP and anti-FLAG antibody. **e** GST alone or GST-LEM4 immobilized on glutathione-agarose beads was incubated with the cell extract of HEK293T cells transfected with GFP-ERα or various mutants of ERα tagged with GFP. Bound proteins were separated by SDS-PAGE and immunoblotted with anti-ERα antibody. **f** MCF7-LEM4 cells were immunostained with anti-FLAG (indicated LEM4, red) and anti-ERα (green) antibody, and counterstained with DAPI (blue). Scale bars, 7.5 μm. **g** MCF7-LEM4 cells were transfected with siRNA of ESR1 or treated with fulvestrant for 6 days. Total cell viability were assessed by SRB assays. Results are the mean ± s. d. of three independent experiments performed in triplicate. Western blot was performed with anti-ERα antibody. **h** Compared the relative mRNA LEM4 level between pre-treatment and post-treatment in samples from GEO GSE33658. ***P < 0.001. Tukey's multiple comparisons test for **g**

choice for different times, cell fixation and SRB staining, and absorbance measurement. The EdU incorporation assay was performed according to the manufacturer's instructions (EdU- assay kit, Beyotime Biotechnology, Shanghai, China). Briefly, cells were cultured in 24-well plates and 50 μM EdU added to each well. The cells were cultured for an additional 2 h. Cells were subsequently fixed on glass coverslips with 4% paraformaldehyde before undergoing Apollo staining for 30 min and Hoechst 33342 staining for 30 min. The EdU incorporation rate was expressed as the ratio of EdU-positive cells to total Hoechst-positive cells. Experiments were performed in triplicate.

For cell cycle analysis, cells were collected and washed twice with ice-cold phosphate-buffered saline (PBS). The cells were fixed with 70% ethanol at 4 °C for 24 h and incubated with 4′,6-diamidino-2-phenylindole (DAPI) (Sigma-Aldrich, St. Louis, Missouri) for 10 min. Samples were analyzed using Millipore Amnis® Imaging Flow Cytometers (EMD Millipore, Darmstadt, Germany).

For cell viability assays, cells were plated at $2 \times 10^4$ cells per well in 24-well plates, in triplicate, in the presence of 5% dextran-charcoal-treated FBS (DCC-FBS) with 4-OHT, PD0332991 treatments at stated concentrations, a combination of the two, or vehicle for 9 days. Cell viability was measured by SRB assay. Triplicates were averaged for mean absorbance, and a percentage calculated for the survival of drug-treated cells versus time-matched vehicle-treated cells. Experiments were performed in triplicate.

**Transwell invasion assay.** The Boyden chamber assay was used for invasion assay. Briefly, $1 \times 10^5$ cells suspended in 200 μl serum-free medium were plated into the top chamber with 50 μl growth factor reduced Matrigel-coated membrane (8 μM pole size, BD Biosciences, Shanghai, China). The chambers were then placed into 24-well plates with 600 μl serum-containing (10%) medium in each well. After 24 h incubation, cells on the bottom side of the chamber membrane were fixed, stained with crystal violet and photographed.

**Soft agar colony formation assay.** The MCF7 and T47D cell lines and their derived cell lines were cultured in DMEM or RPMI-1640/ with 10% FBS in 6-well plates within a 0.35% agar layer, and $2 \times 10^3$ cells were seeded to the middle layer of the soft agar (Lonza, Rockland, USA). The plates were incubated for 14 days (T47D) or 21 days (MCF7), after which the cultures were inspected and photographed. Assays were conducted in triplicate in a single experiment, and then as three independent experiments.

**3D matrigel assay.** Briefly, the 8-well chamber slide (BD Bioscience, catalog number: 354108) is precoated with 80 μl growth factor reduced BD Matrigel$^{TM}$ Matrix (BD Biosciences, catalog number: 354230) per well. Then the chamber slide is transferred to a cell culture incubator to allow matrigel solidification for at least 15 min. Five thousand cells for MCF-10A in assay medium containing 5 ng ml$^{-1}$ EGF and 2% Matrigel were seeded in each well. Medium were replenished every 3 days. Images of spheres with defined scales were subjected to the ImageJ computer program to determine the area covered by each sphere, and the diameter of that sphere was then calculated based on the circle formula.

**Luciferase reporter assay.** MCF7 cells were transiently transfected with EREα-Luc, pCMV6-FLAG-LEM4, or pCMV6-FLAG and pRL-TK (Renilla luciferase, Promega) as an internal control. MDA-MB-231 cells were transiently transfected with EREα-Luc, wild-type ERα, pCMV6-FLAG-LEM4, or pCMV6-FLAG and pRL-TK. After 48 h of transfection, luciferase activity was measured using a luminometer (Tristar LB941, Berthold Technologies, BadWild, Germany). Firefly luciferase activity was normalized to the Renilla luciferase activities. Experiments were performed in triplicate.

**Chromatin immunoprecipitation.** MCF7 cells, MCF7-LEM4 cells, and MCF7-LEM4 cells with depletion of LEM4 expression by siRNA, grown for 3 days in

phenol red-free DMEM supplemented with 5% charcoal-treated FBS. Cells were treated with either vehicle or 10 nM E2 for various time. The Simple ChIP® Enzymatic Chromatin IP Kit (Cell Signaling, catalog number: #9003) was used to perform the chromatin immunoprecipitation (ChIP) assay according to the manufacturer's instruction. Briefly, cells were cross-linked with 1% formaldehyde for 10 min at room temperature. Glycine quenched samples were washed with ice-cold PBS, collected, and then nuclei were collected after cell lysis. Micrococcal nuclease was added to the nuclei suspension to digest the DNA for 20 min at 37 °C. Subsequently, the digest reactions were stopped by the addition of 0.5 M EDTA. Nuclear pellet was collected and were incubated in ChIP buffer with protease inhibitors for 10 min on ice. Sheared cross-linked-chromatin preparation was collected after sonication. Chromatin extracts containing DNA fragments of 150–900 base pairs were immunoprecipitated using anti-ERα or anti-IgG antibody. Quantitative real-time PCR analyses were performed using the Realplex real-time PCR detection system (Eppendorf). The sequences of the primers described in Supplementary Table 4.

**RNA interference.** SiRNA oligos for RNAi against human LEM4, AKT, and Aurora-A were synthesized by Invitrogen (Supplementary Table 1) and transfected using the Lipofectamine RNAiMAX reagent according to the manufacturer's instructions (Invitrogen).

**RNA extraction and real-time RT-PCR.** Total RNA was isolated using Trizol reagent (Roche Diagnostics, Indianapolis, Indiana) according to the manufacturer's protocol. A total of 2 μg of RNA was reverse-transcribed using the PrimeScript™ RT reagent Kit with gDNA Eraser (TaKaRa). The SYBR green (TaKaRa) method was used with the Realplex real-time PCR detection system (Eppendorf) to detect gene expression. Real-time PCR was performed in triplicate. The sequences of the oligonucleotide primers used for real-time PCR are described in Supplementary Table 3.

**Western blot analysis.** The cells were collected and resuspended in cell lysis buffer containing 50 mM Tris (pH 7.4), 150 mM NaCl, 1 mM EDTA, 1 mM EGTA, 1 mM NaF, 1 mM Na3VO4, 1% Triton X-100, 10% glycerol, 0.25% deoxycholate, and 0.1% SDS. Lysates were electrophoresed using SDS-PAGE and blotted onto nitrocellulose (NC) membrane. Membranes were blocked with 5% nonfat milk or 5% BSA solution for 2 h. Samples were probed with primary antibodies overnight at 4 °C (for antibody details, see Supplementary Table 5). Secondary antibodies HRP conjugated donkey anti-Rabbit IgG (GE Healthcare NA934V) or goat anti-mouse IgG (H + L) (ZB2305) were diluted at 1:5000. Blots were photographed by the Image Quant LAS 4000 luminescent image analyzer (General Electric, Fairfield, CT). All Western blots were quantified using the Image J program (NIH, USA). Uncropped scans can be found in Supplementary Fig. 11, 12, 13, 14, 15.

**Immunofluorescence staining.** Cells were fixed with 4% paraformaldehyde for 30 min, permeabilized with 0.5% (vol/vol) Triton X-100 for 20 min and blocked with 10% normal goat serum in phosphate-buffered saline for 30 min at room temperature. Cells were then incubated with primary antibodies (E-cadherin, 1:1000; ERα, 1:100; FLAG, 1:400) overnight at 4 °C. After washing, cells were incubated with secondary antibodies conjugated with FITC (anti-rabbit antibody, 1:1000) or RITC (anti-mouse antibody, 1:500) at room temperature for 2 h and washed three times with PBS. Nuclei were stained with DAPI (4′,6-diamidino-2-phenylindole) for 10 min. The fluorescence images were taken with confocal microscope (Leica TCS SP5, Germany).

**Immunoprecipitation and GST pull-down assay.** For immunoprecipitation, cells were transfected with pCMV6-FLAG-LEM4 or the empty vector and the cell lysate incubated with anti-FLAG-agarose beads at 4 °C for 2 h (Sigma). The beads were washed extensively and eluted under native conditions by competition with 3×FLAG peptide (Sigma). The supernatants were analyzed by Western blotting as

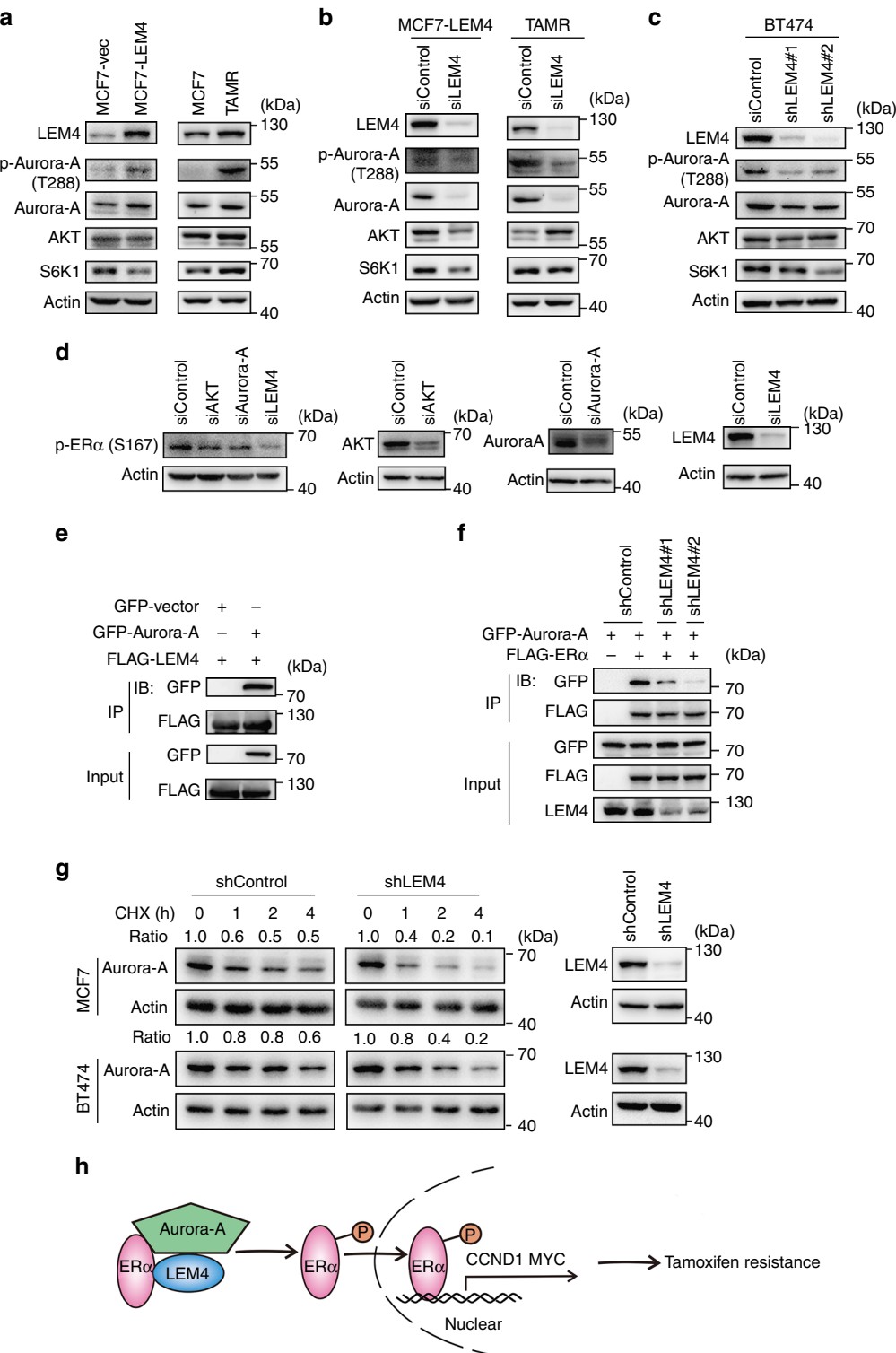

**Fig. 8** LEM4 mediated the phosphorylation of ERα-Ser167 by Aurora-A. **a** Immunoblot analysis of the phosphorylation of Aurora-A, Aurora-A, AKT, and S6K1 in MCF7-LEM4 and MCF7-TAMR cells. **b** LEM4 siRNA-treated and control siRNA-treated MCF7-LEM4 and MCF7-TAMR cells were treated for 48 h. Western blot was performed with indicated antibodies. **c** BT474 cells were transfected with *LEM4* siRNA or control siRNA for 48 h. Western blot was performed with indicated antibodies. **d** MCF7-LEM4 cells were transfected with *LEM4* siRNA or *AKT* siRNA or *Aurora-A* siRNA for 48 h. Western blot was performed with indicated antibodies. **e** HEK293T cells were transfected with FLAG-LEM4 and GFP-Aurora-A. After incubating for 48 h, cell lysates were precipitated with anti-FLAG affinity gel and immunoblotted with anti-GFP antibody. **f** HEK293T-shLEM4 cells were transfected with FLAG-ERα and GFP-Aurora-A. After incubating for 48 h, cell lysates were precipitated with anti-FLAG affinity gel and immunoblotted with anti-GFP antibody. **g** MCF7-shControl and MCF7-shLEM4 cells, or BT474-shControl and BT474-shLEM4 cells were treated with 50 μg mL$^{-1}$ CHX for 0, 1, 2, and 4 h and analyzed for Aurora-A by immunoblot. **h** Model of LEM4 regulation of the ERα signaling leading to tamoxifen resistance in ER+ breast cancer

described above. For GST pull-down assays, the GST alone or recombinant GST-LEM4 immobilized on glutathione-agarose beads was incubated with the MCF7 cell extract at 4 °C for 2 h. The beads were washed extensively and the pulled-down proteins mixed with SDS-PAGE sample buffer for analysis by immunoblotting with appropriate antibodies.

**Mouse xenograft studies**. All animal studies were approved by the Institutional Animal Care and Use Committee at Nankai University. For Fig. 2f, g, estrogen pellets (60-day slow release pellet containing 0.72 mg; Innovative Research of America) were implanted subcutaneously at the nape of the neck of female 3 to 4-week-old BALB/c nu/nu athymic mice (Vitalriver Beijing, China). Three days later, $5 \times 10^6$ MCF7-vec/MCF7-LEM4 or T47D-vec/T47D-shLEM4 cells were suspended in 100 µL of PBS/Matrigel (1:1) (BD Biosciences, Franklin Lakes, New Jersey) and injected subcutaneously. For Supplementary Figure 5b, $5 \times 10^6$ MCF7-vec/MCF7-LEM4 cells were suspended in 100 µL of PBS/Matrigel (1:1) (BD Biosciences) and injected subcutaneously into the nude mice without estrogen pellet implantation. For Fig. 4f, g, $5{\sim}8 \times 10^6$ cells ($5 \times 10^6$ MCF7 cells and $8 \times 10^6$ BT474 cells) suspended in 100 µL of PBS/Matrigel (1:1) were injected subcutaneously into the female nude mice with the estrogen pellets implanted 3 days prior. For MCF7 cells in Fig. 4f, when the tumor size reached ~100 mm³, seven mice in each group were treated with or without tamoxifen pellets implanted subcutaneously (60-day slow release pellet containing 5 mg; Innovative Research of America). For BT474 cells in Fig. 4g, mice were treated with tamoxifen pellets implanted subcutaneously and when the tumor size reached ~200 mm³. For Figs 5c, $1 \times 10^7$ cells suspended in 100 µL of PBS/Matrigel (1:1) were injected subcutaneously into the mammary fat pads of female nude mice with the estrogen pellets implanted 3 days prior. When the tumor size reached ~200 mm³, the mice were randomized into three groups and treated with subcutaneous tamoxifen pellet implants, 100 mg kg⁻¹ PD0332991 (MedChem Express) tricubic weekly by intragastric-administration for 6 weeks, or a combination of tamoxifen and PD0332991. After 8 to 10 weeks, mice were euthanized and the tumors were sectioned and fast-frozen, or formalin-fixed and paraffin-embedded, and H&E-stained slides made. The tumors were measured using a Vernier caliper and the tumor volume was estimated as follows: V = (length × width × height × 0.5) mm³.

**IHC in xenograft tumors**. Tissue sections of xenograft tumor (4-µm-thick) were immunostained with anti-LEM4 (1:500), anti-Ki-67 (1:500), anti-p-CDK1 (1:500), anti-p-CDK4 (1:500), anti-cyclin D1 (1:2000), anti-p-Rb (1:500), anti-caspase 3/7 (1:500), or anti-p-ERα167 (1:500) antibodies overnight at 4 °C. The peroxidase-conjugated streptavidin complex method was performed, followed by the 3, 3′ diaminobenzidine (DAB) procedure according to the manufacturer's protocols (Dako, Agilent pathology solutions).

**IHC for tissue microarrays**. Breast cancer tissue microarray slides (HBre-Duc150Sur-01/02) were purchased from Shanghai Outdo Biotech Co. Ltd. (SOBC). All patients gave consent for the use of their tissue samples and clinical data. The ethic review board at the Tianjin Central Hospital of Gynecology Obstetrics approved the study. Staining of the tissue microarray was performed according to previously described protocols[60]. Briefly, after deparaffinization in xylene and rehydration in a series of alcohols (100–75%), slides were incubated in the dual endogenous enzyme block (Dako, Carpinteria, CA) for 15 min to inactivate endogenous peroxide activity and treated in citrate buffer (pH 6.0) for 3 min in a pressure cooker for antigen retrieval. After cooling for 45 min at room temperature, slides were incubated with rabbit anti-LEM4 antibody at 1:400 dilution at 4 °C overnight and with the secondary horseradish peroxidase-labeled polymer anti-rabbit IgG for 30 min at room temperature. With diaminobenzidine tetra-hydrochloride (Dako) as a chromogen, slides were counter stained with hema-toxylin. Slides were scanned at 20× using the Aperio ScanScope XT pathology system (Leica Microsystems, Germany). Images were exported to Aperio Image-Scope for viewing. The LEM4 staining score was calculated according to the following procedures. The intensity of staining (0 = negative, 1 = low, 2 = medium, and 3 = high) and the percentage of positively stained cells (0–100%) were recorded for each specimen, and the LEM4 expression score was calculated as intensity score × percentage of positive cells × 100. In order to translate continuous LEM4 expression into a clinical decision, the median cutoff point was used to stratify patients into two groups.

**Statistical analysis**. All in vitro experiments were repeated at least three times unless stated otherwise. Student's t-test or one-way ANOVA for multiple group comparisons were performed using SPSS 21. Survival curves were plotted using Kaplan–Meier analysis and the statistical parameters calculated by a log-rank test using GraphPad Prism 6 software. All statistical tests were two-sided, and differences were considered statistically significant at $P < 0.05$ unless stated otherwise.

## Data availability
The data that support the findings of this study are available from the corresponding author upon reasonable request. The URL was provided for each of GEO datasets, which was obtained from Gene Expression Omnibus (GEO) database (https://www.ncbi.nlm.nih.gov/geo). GSE2990, GSE3494, GSE9195, GSE33658, GSE2034, GSE16446, GSE20685, GSE100075.

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

## Acknowledgements

The authors thank Drs. Yushan Zhu and Jun Zhou for providing plasmids and materials. The authors thank Dr Tao Zhu for providing MCF7-TAMR cell model. The study was supported by the National Natural Science Foundation of China (Grant No. 91649107 and 81402320), the Natural Science Foundation of Tianjin City of China (Grant No. 17JCYBJC24100 and 16JCYBJC27000), and the Strategic Priority Research Program (Pilot study) "Biological basis of aging and therapeutic strategies" of the Chinese Academy of Sciences (grant XDPB10).

## Author contributions

Z.Z., J.T.D., and J.H. conceived and designed the study. A.G., T.S., and G.M. performed cell culture experiments, histology, immunohistochemistry, and analysis. A.G. and J.C. performed Mouse xenograft studies. Z.Z. and L.C. contributed tissue array experiments and recording data analysis. Z.Z. and Y.W. performed Kaplan–Meier analysis. A.G. and Q.W. performed cell cycle analysis. Q.H., T.S., Q.W., and J.S. contributed the GST-pulldown and IP and results. A.G. and R.W. contributed CHX chase assay and results. J. Z., L.L., J.T.D. provided comments on the manuscript. Z.Z., J.H., and A.G. contributed manuscript preparation.

## Additional information

**Competing interests:** The authors declare no competing interests.

