## [Peer Review File · Nature Communications]

Reviewers' comments:

Reviewer #1 (Remarks to the Author):

This manuscript identifies nuclear envelope related proteins in breast cancer, which revealed LEM4 as a factor expressed in tumor, but less so in adjacent normal. LEM4 is shown to correlate with poor clinical outcome, regardless of ER+ subtype. LEM4 overexpression is shown to promote cell growth in a number of cancer cell lines both in vitro and as colonies. LEM4 also promoted tumor growth when implanted as xenografts. Key cell cycle genes are shown to be influenced by LEM4 expression and Palbociclib can reverse the tamoxifen-resistance mediated by LEM4 overexpression.

This is a thorough and well-written manuscript. The findings are convincing and of relevance. I have several points that need to be addressed, but if these can be addressed, this would be an excellent contribution to the community.

- LEM4 overexpression appears to promote growth of all cancer cells, including ER+ and ER-. Does LEM4 overexpression in a normal cell line also promote cell growth?
- LEM4 is shown to make ER+ cancer cells estrogen independent, but does it make cells ER independent? Would siRNA to ER or Fulvestrant treatment reverse the LEM4-mediated growth of ER+ cancer cells?
- Why are the cell cycle genes shown to be upregulated when LEM4 is overexpression MCF7 cells, but down-regulated when LEM4 is inhibited in T47D cells? Why are these experiments not performed in the same cell lines?
- If LEM4 can interact with CDK4 and pRb and it can also interact with ER, why have CDK4-ER interactions not been seen in any unbiased IP-Mass Spec experiments?
- The use of ER-Luc vectors to assess ER activity is not appropriate. These are the most rudimentary of assays, they lack chromatin context and it is well established that ER does not typically regulate from promoters. Can LEM4 change ER binding as assessed by ChIP-qPCR assays?

Reviewer #2 (Remarks to the Author):

LEM4 confers tamoxifen resistance to breast cancer cells through activation of the cyclin D/CDK4/6/Rb axis and ER α signaling

LEM4 is required for the function of both a mitotic kinase and a phosphatase that act on BAF, an essential effector of nuclear assembly. LEM4 is required for dephosphorylation of BAF by inhibiting its mitotic kinase, VRK-1. LEM4 also interacts with PP2A and is required for it to dephosphorylate BAF during mitotic exit. By coordinating VRK-1 and PP2A mediated signaling on BAF, LEM4 controls post mitotic NE formation. In the following manuscript Gao and colleagues provide evidence for the role of LEM4 in resistance to tamoxifen. They show LEM4 expression appears higher in luminal B, HER2+ and basal breast cancer. Ectopic expression of LEM4 led to estrogen independent tumour growth in xenograft models and associated with tamoxifen resistance whilst knockdown of LEM4 showed the opposite. The authors suggest LEM4 interacts with ER and aurora kinase leading to phosphorylation of ERser167 and transcription of CCND1 and cmyc. They provide evidence that LEM4 stabilises CDK4 and promotes phosphorylation of RB. Finally the authors suggest cross-talk between ER and LEM4 can be targeted with CDK4/6 inhibitor palbociclib.

Whilst of some interest the data is over-stated particularly the clinical interpretation. Furthermore, the cell lines used are not optimal for this analysis. The authors state in their introduction line 83 MCF7-LEM4 cells "acquired resistance" to tamoxifen. This is not the case, over-expression of LEM4 appeared to make the cell line ligand-independent i.e de-novo this is not an acquired resistant

model.

Major points clinical

1. Analysis of breast cancer sub-types suggests LEM4 is a marker of proliferation. Hence expression is more strongly associated with basal sub-type. The analysis of the clinical data is flawed in multiple ways (fig 1 and suppl S1). The analysis of the clinical data uses "best-cut-off" in order to reflect the data in the best possible light to support the author's hypotheses. The data should be analysed using median cut-off in each case rather than inconsistent cut points. The time points also vary from 250 months to 400 months. Furthermore, publically available data from patients treated in short-term presurgical studies with aromatase inhibitors and fulvestrant are available. Baseline expression and on treatment expression of LEM4 could be correlated with ki67 change or residual ki67.

.

In vitro

2. The authors show that over expression of LEM4 in MCF7 and T47D cells leads to ligand-independent proliferation in xenografts and in 2D culture, they then suggest this is associated with tamoxifen resistance when in fact it appears to be associated with de novo resistance to E-deprivation modeling resistance to an aromatase inhibitor as well as tamoxifen. Again there is sufficient publically available data from LTED models to assess if LEM4 is associated with the LTED phenotype. Whilst the experiments carried out within the manuscript as a whole are logical they are also largely correlative relying on LEM4 over-expression and depletion in-vitro and in vivo. Whilst I realize the authors have used BT474 as a model of tamoxifen resistance, it does not adequately address "acquired" resistance. Furthermore it is HER2 over-expression driving downstream signaling that leads to ligand independent ER activity and resistance to tamoxifen in this model. HER2 activation of ERK1/2 has been shown to alter the balance of coactivators and corepressors leading to the resistant phenotype. Moreover, HER2 regulates emerin's ability to bind BAF raising further questions. Does siRNA knockdown of LEM4 in BT474 alter the expression of HER2 and conversely does knockdown of HER2 impede tamoxifen resistance and does this impact on LEM4 expression.

3. Supporting evidence for the role of LEM4 in TAMR and LTED models for the key experiments would be helpful and would validate their argument.

4. The authors suggest LEM4 leads to tumour "firmly attached to surrounding tissue" what does this mean are there alterations in expression of adhesion molecules as a result of LEM4 over expression or are they implying they are invading the surrounding tissue? This is important and needs clarity.

5. The authors, later in the manuscript, provide evidence that LEM4 over-expression stabilizes ER. This leads to two questions. Firstly does over-expression of LEM4 confer resistance to fulvestrant (ICI182780) or indeed siRNA knockdown of ESR1. If so this would suggest it is an ER-independent not dependent mechanism. Secondly as LEM4 is suggested to stabilize ER does this alter its interaction time with ERE on target genes. To test this a ChIP-timecourse investigating ER and CBP recruitment/cycling to E-regulated gene promoters (TFF1, PGR, GREB1, and CCND1) should be explored in MCF7-LEM4 versus MCF7 control versus siRNA knockdown of -LEM4 in MCF7-LEM4 to assess phenotype loss and rescue. A Re-ChIP for LEM4 would also allow a conclusion to be drawn regarding its role in potentiating ER transcriptional activity and indeed its interaction with ER. Whilst I appreciate the author's show with GST-ER truncations that LEM4 binds to ER's LBD, this is an artificial system. As the authors show LEM4 is required for tamoxifen resistance in BT474, a pulldown of ER with LEM4 in a "native" rather than over-expressed background would be helpful. Furthermore confocal microscopy to show co-localisation would be useful.

6. With regard to figure 4 A and C I am unclear if E2 was added to these assays, The IC50 for wt-MCF7 in response to 4-OHT appears high, in fact there is only a c35% drop in proliferation

between 1nM-1000nM and both MCF7 and MCF7-LEM4 show zero viability at 10uM. This is not a characteristic response curve for MCF7 cells, which generally show an IC50 of 10nM for 4-OHT in the presence of E2. Also the scales on 4A and C show different units.

7. Whilst the authors suggest ER is stabilized the WB evidence is less than compelling and it is pERser167, which appears the main feature that is altered.

8. The over expression and knockdown assays switch from cell line to cell line. Whilst I appreciate that confirmation in multiple cell lines is necessary, it would be reassuring to see the MCF7 control and MCF7-LEM4 models used for all key experiments Fig 2, fig 6 A, fig 6f so that LEM4 is depleted to give a direct comparison rather than versus BT474.

9. One of my major concerns is the data shown in Fig 3G suggesting that LEM4 influences G1/S transition. Both T47D and MCF7 are well documented as p16 null. Perhaps the models should be checked.

10. The authors suggest that LEM4 over expression stabilizes CDK4, RB and ER. However, as LEM4 also coordinates the control of BAF dephosphorylation by inhibiting VRK1 supporting PP2A action on BAF it would be good as a control to assess the impact on the more classical role of LEM4.

11. In the discussion the authors state LEM4 over-expression results in abnormal chromosome segregation in late metaphase and anaphase but do not show the data, this is important. Furthermore, page 13 they suggest p16 inactivates CDK4 and that p16 abundance is reduced in MCF7-LEM4 yet MCF7 and T47D are p16 null cells hence their sensitivity to CDK4/6 inhibitor as shown by Finn et al.

12. Overall, whilst of some interest, the study does not provide any key data of clinical relevance the fact that CDK4/6 inhibitors show antiproliferative activity in ER+ BC in both the primary metastatic and after progression on endocrine therapy is well document. As regards LEM4 expression and association with poor response to therapy – this need further exploration in datasets described.

Other points

Whilst I understand that space constraints maybe an issue the methodology is hard to follow and needs to be more detailed.

Legend on Figure S1 B should read luminal A

Reviewer #1 (Remarks to the Author):

This is a thorough and well-written manuscript. The findings are convincing and of relevance. I have several points that need to be addressed, but if these can be addressed, this would be an excellent contribution to the community.

- LEM4 overexpression appears to promote growth of all cancer cells, including ER+ and ER-. Does LEM4 overexpression in a normal cell line also promote cell growth?

We thank the reviewer for this suggestion. We attempted to generate MCF-10A cells expressing LEM4 stably but we were not able to obtain the cell lines in due time. We then knocked down LEM4 by transfecting siRNAs into MCF-10A cells and measured the cell proliferation. Depletion of LEM4 in MCF-10A cells resulted in significantly decreased cell growth in monolayer culture (Fig. S2D). We also observed that interference with LEM4 expression would result in a significant inhibition of mammosphere formation of MCF10A cells in Matrigel (Fig. S2E). We will continue the trials on making LEM4 transformed normal mammary epithelial cells (MCF10A-LEM4 cells). If we obtained the results after the revision is evaluated, we will add this part.

- LEM4 is shown to make ER+ cancer cells estrogen independent, but does it make cells ER independent? Would siRNA to ER or Fulvestrant treatment reverse the LEM4-mediated growth of ER+ cancer cells?

We thank the reviewer for this suggestion. We performed the experiments as suggested and found that ER α downregulation with fulvestrant or siRNA inhibited estrogen-independent growth of MCF7-LEM4 cells (Fig. 7G).

- Why are the cell cycle genes shown to be upregulated when LEM4 is overexpression MCF7 cells, but down-regulated when LEM4 is inhibited in T47D cells? Why are these experiments not performed in the same cell lines?

We thank the reviewer for pointing this out. Our findings suggest that LEM4 overexpression accelerates tumor growth. Tumor development is highly related to uncontrolled cell growth. LEM4 alters cell cycle distribution by promoting the G1 to S phase transition. Moreover, LEM4 could physically interact with and stabilize ER α , and subsequently promote ER α transcriptional activity which resulted in upregulating both cyclin D1 and c-Myc. Thus, the cell cycle genes shown to be upregulated when LEM4 is overexpression the tumor cells.

We have performed the experiments in the same cell lines (Fig. S6F, S6G).

- If LEM4 can interact with CDK4 and pRb and it can also interact with ER, why have CDK4-ER interactions not been seen in any unbiased IP-Mass Spec experiments?

LEM4 protein localized to the endoplasmic reticulum and the inner nuclear membrane. LEM4 has ankyrin repeat domain that is one of the most common protein-protein interaction motifs in nature. Thus, there is a possibility that LEM4 may form diverse group of temporal or spatial protein complexes. In our manuscript, we did not observed CDK4-ER interactions also.

- The use of ER-Luc vectors to assess ER activity is not appropriate. These are the most rudimentary of assays, they lack chromatin context and it is well established that ER does not typically regulate from promoters. Can LEM4 change ER binding as assessed by ChIP-qPCR assays?

We thank the reviewer for this suggestion. We performed the experiments as suggested and found the occupancy of ER α to the ER α -binding sites was significantly enhanced in MCF7-LEM4 cells (Fig. 6D).

Reviewer #2 (Remarks to the Author):

LEM4 confers tamoxifen resistance to breast cancer cells through activation of the cyclin D/CDK4/6/Rb axis and ER α signaling

LEM4 is required for the function of both a mitotic kinase and a phosphatase that act on BAF, an essential effector of nuclear assembly. LEM4 is required for dephosphorylation of BAF by inhibiting its mitotic kinase, VRK-1. LEM4 also interacts with PP2A and is required for it to dephosphorylate BAF during mitotic exit. By coordinating VRK-1 and PP2A mediated signaling on BAF, LEM4 controls post mitotic NE formation. In the following manuscript Gao and colleagues provide evidence for the role of LEM4 in resistance to tamoxifen. They show LEM4 expression appears higher in luminal B, HER2+ and basal breast cancer. Ectopic expression of LEM4 led to estrogen independent tumour growth in xenograft models and associated with tamoxifen resistance whilst knockdown of LEM4 showed the opposite. The authors suggest LEM4 interacts with ER and aurora kinase leading to phosphorylation of ERser167 and transcription of CCND1 and cmyc. They provide evidence that LEM4 stabilizes CDK4 and promotes phosphorylation of RB. Finally the authors suggest cross-talk between ER and LEM4 can be targeted with CDK4/6 inhibitor palbociclib.

Whilst of some interest the data is over-stated particularly the clinical interpretation. Furthermore, the cell lines used are not optimal for this analysis. The authors state in their introduction line 83 MCF7-LEM4 cells “acquired resistance” to tamoxifen. This is not the case, over-expression of LEM4 appeared to make the cell line ligand-independent i.e de-novo this is not an acquired resistant model.

We have modified the sentences accordingly.

Major points

clinical1. Analysis of breast cancer sub-types suggests LEM4 is a marker of proliferation. Hence expression is more strongly associated with basal sub-type. The analysis of the clinical data is flawed in multiple ways (fig 1 and suppl S1). The analysis of the clinical data uses “best-cut-off” in order to reflect the data in the best possible light to support the author’s hypotheses The data should be analysed using median cut-off in each case rather than inconsistent cut points. The time points also vary from 250 months to 400 months. Furthermore, publically available data from patients treated in short-term presurgical studies with aromatase inhibitors and fulvestrant are available. Baseline expression and on treatment expression of LEM4 could be correlated with ki67 change or residual ki67.

We thank the reviewer for this suggestion. We have performed the experiment as suggested. The patients were separated into two groups according to the median value of LEM4 protein expression. The results showed that the association between a higher expression of LEM4 and a poorer survival rate of patients was statistically significant (Fig. 1E, 1F, 1G, 1H).

With regard to the question of time variation from 250 months to 400 months, we combined all the microarrays datasets in KM database (www.kmplot.com) in our previous analysis. This method would give a good advantage of large sample size. We found that the method to combine the microarrays datasets was applied to the KM survival analysis among other studies (www.kmplot.com), (for example, (Oncogene, 2014;33(42):4985-4996) and (Nat Communications. 2015;6:8746). However, to enable to measure the LEM4 gene with the same sensitivity, specificity and dynamic range, the datasets from the online database were performed to the KM survival analysis each separately. Here, we showed the results from the dataset GSE2034, GSE16446, GSE20685 and GSE2990 (Fig. 1I).

We performed the Pearson's correlation analysis between *LEM4* and *MKI67* mRNA levels from the datasets GSE2990 (Gene Expression Profiling in Breast Cancer: Understanding the Molecular Basis of Histologic Grade To Improve Prognosis). A positive correlation between *LEM4* and *MKI67* was observed at the mRNA level (Fig. S3).

In vitro

2. The authors show that over expression of LEM4 in MCF7 and T47D cells leads to ligand-independent proliferation in xenografts and in 2D culture, they then suggest this is associated with tamoxifen resistance when in fact it appears to be associated with de novo resistance to E-deprivation modeling resistance to an aromatase inhibitor as well as tamoxifen. Again there is sufficient publically available data from LTED models to assess if LEM4 is associated with the LTED phenotype. Whilst the experiments carried out within the manuscript as a whole are logical they are also largely correlative relying on LEM4 over-expression and depletion in-vitro and in vivo. Whilst I realize the authors have used BT474 as a model of tamoxifen resistance, it does not adequately address "acquired" resistance. Furthermore it is HER2 over-expression driving downstream signaling that leads to ligand independent ER activity and resistance to tamoxifen in this model. HER2 activation of ERK1/2 has been shown to alter the balance of coactivators and corepressors leading to the resistant phenotype. Moreover, HER2 regulates emerin's ability to bind BAF raising further questions. Does siRNA knockdown of LEM4 in BT474 alter the expression of HER2 and conversely does knockdown of HER2 impede tamoxifen resistance and does this impact on LEM4 expression.

We thank the reviewer for this suggestion. We performed statistical analyses on the RNAseq GEO dataset GSE100075 from LTED models (long-term estrogen deprivation in MCF7 cell line with and without ESR1 mutations) to assess if LEM4 is associated with the LTED phenotype. The results revealed that LEM4 expression was significantly elevated in both MCF7_LTED_ESR1_WT and MCF7_LTED_ESR1_Y537C models (Fig. 4B).

We showed that siRNA knockdown of LEM4 in BT474 did not alter the expression of HER2. Conversely, the LEM4 levels was decreased upon knockdown of HER2 in BT474 cells. Moreover, knockdown of HER2 impeded tamoxifen resistance in BT474 cells (Fig. S7A, S7B).

3. Supporting evidence for the role of LEM4 in TAMR and LTED models for the key experiments would be helpful and would validate their argument.

We thank the reviewer for this suggestion. We have performed the experiments in the MCF7-TAMR cell model. The results supported that LEM4 renders cells resistant to tamoxifen (Fig. 4A, 4C, 5A, 5B, 6A, 6B, 8A, 8B). We have modified the text accordingly.

The MCF7-TAMR cell model was kindly provided by Dr Tao Zhu (University of Science and Technology of China).

4. The authors suggest LEM4 leads to tumour “firmly attached to surrounding tissue” what does this mean are there alterations in expression of adhesion molecules as a result of LEM4 over expression or are they implying they are invading the surrounding tissue? This is important and needs clarity.

We thank the reviewer for this suggestion. The MCF7-LEM4 cells were subjected to migration and invasion assays, and the results showed that overexpression of LEM4 in MCF7 cells leads to significant increase of cell migration and invasion. Real-time PCR analysis and immunoblots of EMT markers revealed that LEM4 overexpression caused elevated expression of Slug and ZEB1. Moreover, immunostaining of MCF7-LEM4 cells using antibody (anti E-cadherin) showed the loss of E-cadherin in cell-cell contacts (Fig. S4A, S4B, S4C, S4D).

5. The authors, later in the manuscript, provide evidence that LEM4 over-expression stabilizes ER. This leads to two questions. Firstly does over-expression of LEM4 confer resistance to fulvestrant (ICI182780) or indeed siRNA knockdown of ESR1. If so this would suggest it is an ER-independent not dependent mechanism. Secondly as LEM4 is suggested to stabilize ER does this alter its interaction time with ERE on target genes. To test this a CHIP-timecourse investigating ER and CBP recruitment/cycling to E-regulated gene

promoters (TFF1, PGR, GREB1, and CCND1) should be explored in MCF7-LEM4 versus MCF7 control versus siRNA knockdown of -LEM4 in MCF7-LEM4 to assess phenotype loss and rescue. A Re-ChIP for LEM4 would also allow a conclusion to be drawn regarding its role in potentiating ER transcriptional activity and indeed its interaction with ER. Whilst I appreciate the author's show with GST-ER truncations that LEM4 binds to ER's LBD, this is an artificial system. As the authors show LEM4 is required for tamoxifen resistance in BT474, a pulldown of ER with LEM4 in a "native" rather than over-expressed background would be helpful. Furthermore confocal microscopy to show co-localisation would be useful.

We thank the reviewer for these suggestions.

We performed the experiments as suggested and found that ER α downregulation with fulvestrant or siRNA inhibited estrogen-independent growth of MCF7-LEM4 cells (Fig. 7G).

We performed the ER α ChIP and time course ER α ChIP as suggested and found the LEM4 enhanced the ER binding with estrogen response element (ERE) on target genes compared with the control, and silencing LEM4 by RNAi in MCF7-LEM4 cells reverses these events (Fig. 6D, 6E).

LEM4 was detected in ER α immunoprecipitates in BT474 cells (Fig. 7C).

Due to the anti-LEM4 antibody is not suitable for immunofluorescence, we performed immunofluorescence assays in MCF7-LEM4 (FLAG-tagged) cells with anti-ER α and anti-FLAG antibodies. The result revealed that ER α co-localized with LEM4 not only in the nuclear envelope but also in cytoplasm (Fig. 7F).

6. With regard to figure 4 A and C I am unclear if E2 was added to these assays, The IC50 for WT-MCF7 in response to 4-OHT appears high, in fact there is only a c35% drop in proliferation between 1nM-1000nM and both MCF7 and MCF7-LEM4 show zero viability at 10uM. This is not a characteristic response curve for MCF7 cells, which generally show an IC50 of 10nM for 4-OHT in the presence of E2. Also the scales on 4A and C show different units.

We thank the reviewer for pointing this out. We did not add E2 in our previous experiments. Here, we have repeated the IC50 for MCF7 or BT474 cells in response to 4-OHT in the presence of E2, and we showed the scales the same units (Now Fig. 4D, 4E).

7. Whilst the authors suggest ER is stabilized the WB evidence is less than compelling and it is pERser167, which appears the main feature that is altered.

We agree with what the reviewer commented on the ER stability and phosphorylation of ER α -Ser167. In order to clarify this point, we knocked-down LEM4 with LEM4 siRNA in MCF7-LEM4 cells and MCF7-TAMR cells. Remarkably, abrogation of LEM4 expression by siRNA resulted in a depleted abundance of pERser167, concurrently, we observed that a decreased expression of ER α occurred (Fig. 6A and 6B). Taking together with other findings, we concluded that LEM4 activates ER α signaling via regulation both phosphorylation of ER α -Ser167 and ER α stability.

8. The over expression and knockdown assays switch from cell line to cell line. Whilst I appreciate that confirmation in multiple cell lines is necessary, it would be reassuring to see the MCF7 control and MCF7-LEM4 models used for all key experiments Fig 2, fig 6 A, fig 6f so that LEM4 is depleted to give a direct comparison rather than versus BT474.

We thank the reviewer for this suggestion. We have performed the experiments as suggested (Figure. 2C, 6A, 6B, 8A and 8B).

9. One of my major concerns is the data shown in Fig 3G suggesting that LEM4 influences G1/S transition. Both T47D and MCF7 are well documented as p16 null. Perhaps the models should be checked.

We thank the reviewer for pointing this out. We have checked the cell models, and the Cell Line Authentication Reports showed that all the cell lines had a correct short tandem repeats (STR) profiling (Supplementary data the Cell Line Authentication Reports). Further, Immunoblot analysis was performed with anti-p16 monoclonal antibody (BD, 551153) instead of the anti-p16 antibody (proteintech, 10883-1-AP). The positive control is the whole HEK293T cell lysates. The western blot performed with the anti-p16 antibody (BD, 551153) showed that both T47D and MCF7 cells do not express p16. Thus, we removed the associated panels in Fig 3F and 3G.

10. The authors suggest that LEM4 over expression stabilizes CDK4, RB and ER. However, as LEM4 also coordinates the control of BAF dephosphorylation by inhibiting VRK1 supporting PP2A action on BAF it would be good as a control to assess the impact on the more classical role of LEM4.

LEM4 controls postmitotic nuclear envelope formation. CDK4-Rb axis has a pivotal role in the G1/S phase transition. We agree that it would be good as a control to assess the impact on the more classical role of LEM4. We immunoblotted VRK1 and PP2A-C in the LEM4-depleted MCF7 cells. The results revealed that VRK1 decreased but PP2A-C elevated (Fig. S10).

11. In the discussion the authors state LEM4 over-expression results in abnormal chromosome segregation in late metaphase and anaphase but do not show the data, this is important. Furthermore, page 13 they suggest p16 inactivates CDK4 and that p16 abundance is reduced in MCF7-LEM4 yet MCF7 and T47D are p16 null cells hence their sensitivity to CDK4/6 inhibitor as shown by Finn et al.

We have observed that LEM4 overexpression results in abnormal chromosome segregation in late metaphase and anaphase. We do believe this is important. We now focus on the functions of the interaction of LEM4 proteins with mitotic kinases (CDK1, Aurora-A and Aurora-B) and try to uncover how LEM4 orders mitotic events. Generally, the functions of LEM4 on genomic instability are of great interest but require a significant amount of work beyond the scope of the current story.

We thank the reviewer for pointing out the question about p16. We have removed the sentence, depending on the data in response to the 9th major point.

12. Overall, whilst of some interest, the study does not provide any key data of clinical relevance the fact that CDK4/6 inhibitors show antiproliferative activity in ER+ BC in both the primary metastatic and after progression on endocrine therapy is well document. As regards LEM4 expression and association with poor response to therapy – this need further exploration in datasets described.

We thank the reviewer for pointing this out. The exploration in datasets were shown in Fig. 1I, Fig. 4H and Fig. 7H.

Other points

Whilst I understand that space constraints maybe an issue the methodology is hard to follow and needs to be more detailed.

Legend on Figure S1 B should read luminal A

We thank the reviewer for pointing this out. We have detailed the methods and corrected the mistakes.

REVIEWERS' COMMENTS:

Reviewer #1 (Remarks to the Author):

The authors have addressed my concerns.

Reviewer #2 (Remarks to the Author):

I would like to thank Dr Zhengmao Zhu et al. for their response to my comments and am happy with the modifications they have made. An interesting study.